# LRP1 regulates peroxisome biogenesis and cholesterol homeostasis in oligodendrocytes and is required for proper CNS myelin development and repair

Jing-Ping Lin[1], Yevgeniya A Mironova[2], Peter Shrager[3], Roman J Giger[1,2,4,5]*

[1]Department of Cell and Developmental Biology, University of Michigan School of Medicine, Ann Arbor, MI, United States; [2]Cellular and Molecular Biology Graduate Program, University of Michigan Medical School, Ann Arbor, MI, United States; [3]Department of Neuroscience, University of Rochester Medical Center, Rochester, NY, United States ; [4]Department of Neurology, University of Michigan Medical School, Ann Arbor, MI, United States; [5]Interdepartmental Neuroscience Graduate Program, University of Michigan Medical School, Ann Arbor, MI, United States

*For correspondence:
rgiger@umich.edu

Competing interests: The authors declare that no competing interests exist.

**Abstract** Low-density lipoprotein receptor-related protein-1 (LRP1) is a large endocytic and signaling molecule broadly expressed by neurons and glia. In adult mice, global inducible (*Lrp1^{flox/flox};CAG-CreER*) or oligodendrocyte (OL)-lineage specific ablation (*Lrp1^{flox/flox};Pdgfra-CreER*) of *Lrp1* attenuates repair of damaged white matter. In oligodendrocyte progenitor cells (OPCs), *Lrp1* is required for cholesterol homeostasis and differentiation into mature OLs. *Lrp1*-deficient OPC/OLs show a strong increase in the sterol-regulatory element-binding protein-2 yet are unable to maintain normal cholesterol levels, suggesting more global metabolic deficits. Mechanistic studies revealed a decrease in peroxisomal biogenesis factor-2 and fewer peroxisomes in OL processes. Treatment of *Lrp1^{−/−}* OPCs with cholesterol or activation of peroxisome proliferator-activated receptor-γ with pioglitazone alone is not sufficient to promote differentiation; however, when combined, cholesterol and pioglitazone enhance OPC differentiation into mature OLs. Collectively, our studies reveal a novel role for *Lrp1* in peroxisome biogenesis, lipid homeostasis, and OPC differentiation during white matter development and repair.
DOI: https://doi.org/10.7554/eLife.30498.001

## Introduction

In the central nervous system (CNS), the myelin-producing cell is the oligodendrocyte (OL). Mature OLs arise from oligodendrocyte progenitor cells (OPCs), a highly migratory pluripotent cell type (*Rowitch and Kriegstein, 2010*; *Zuchero and Barres, 2013*). OPCs that commit to differentiate along the OL-lineage undergo a tightly regulated process of maturation, membrane expansion, and axon myelination (*Emery et al., 2009*; *Hernandez and Casaccia, 2015*; *Li and Yao, 2012*; *Simons and Lyons, 2013*). Even after developmental myelination is completed, many OPCs persist as stable CNS resident cells that participate in normal myelin turnover and white matter repair following injury or disease (*Fancy et al., 2011*; *Franklin and Ffrench-Constant, 2008*).

LRP1 is a member of the LDL receptor family with prominent functions in endocytosis, lipid metabolism, energy homeostasis, and signal transduction (*Boucher and Herz, 2011*). *Lrp1* is broadly expressed in the CNS and abundantly found in OPCs (*Auderset et al., 2016*; *Zhang et al., 2014*).

Global deletion of *Lrp1* is embryonically lethal (*Herz et al., 1992*) and conditional deletion revealed numerous tissue specific functions in neural and non-neural cell types (*Lillis et al., 2008*). In the PNS, *Lrp1* regulates Schwann cell survival, myelin thickness, and morphology of Remak bundles (*Campana et al., 2006*; *Mantuano et al., 2010*; *Orita et al., 2013*). In the CNS, *Lrp1* influences neural stem cell proliferation (*Auderset et al., 2016*), synaptic strength (*Gan et al., 2014*; *Nakajima et al., 2013*), axonal regeneration (*Landowski et al., 2016*; *Stiles et al., 2013*; *Yoon et al., 2013*), and clearance of amyloid beta (*Kanekiyo and Bu, 2014*; *Kim et al., 2014*; *Liu et al., 2010*; *Zlokovic et al., 2010*). Recent evidence shows that neurospheres deficient for *Lrp1* produce more GFAP$^+$ astrocytes at the expense of O4$^+$ OLs and TuJ1$^+$ neurons (*Hennen et al., 2013*; *Safina et al., 2016*). Whether LRP1 is required for proper CNS myelinogenesis, nerve conduction, or repair of damaged adult CNS white matter, however, has not been examined. Moreover, the molecular basis of how LRP1 influences OPC differentiation remains poorly understood.

LRP1 is a large type-1 membrane protein comprised of a ligand binding 515 kDa α chain noncovalently linked to an 85 kDa β chain that contains the transmembrane domain and cytoplasmic portion. Through its α chain, LRP1 binds over 40 different ligands with diverse biological functions (*Fernandez-Castaneda et al., 2013*; *Lillis et al., 2008*). LRP1 mediates endocytotic clearance of a multitude of extracellular ligands (*May et al., 2003*; *Tao et al., 2016*) and participates in cell signaling, including activation of the Ras/MAPK and AKT pathways (*Fuentealba et al., 2009*; *Martin et al., 2008*; *Muratoglu et al., 2010*). The LRP1β chain can be processed by γ-secretase and translocate to the nucleus where it associates with transcription factors to regulate gene expression (*Carter, 2007*; *May et al., 2002*).

Here, we combine conditional *Lrp1* gene ablation with ultrastructural and electrophysiological approaches to show that *Lrp1* is important for myelin development, nerve conduction, and adult CNS white matter repair. Gene expression analysis in *Lrp1*-deficient OPCs identified a reduction in peroxisomal gene products. We show that *Lrp1* deficiency decreases production of peroxisomal proteins and disrupts cholesterol homeostasis. Mechanistic studies uncover a novel role for *Lrp1* in PPARγ-mediated OPC differentiation, peroxisome biogenesis, and CNS myelination.

## Results

### In adult mice, inducible ablation of *Lrp1* attenuates CNS white matter repair

To study the role of *Lrp1* in CNS myelin repair, we pursued a mouse genetic approach. *Lrp1* global knockout through the germline results in embryonic lethality (*Herz et al., 1992*). To circumvent this limitation, we generated *Lrp1$^{flox/flox}$;CAG-CreER$^{TM}$* mice (*Lrp1* iKO) that allow tamoxifen (TM)-inducible global gene ablation. As control, *Lrp1* mice harboring at least one wild-type or non-recombined *Lrp1* allele were injected with TM and processed in parallel (*Figure 1—figure supplement 1*). TM injection into P56 mice resulted in an approximately 50% decrease of LRP1 in brain without noticeable impact on white matter structure (*Figure 1—figure supplement 1*). One month after TM treatment, *Lrp1* iKO and control mice were subjected to unilateral injection of 1% lysophosphatidylcholine (LPC) into the corpus callosum. The contralateral side was injected with isotonic saline (PBS) and served as control. Brains were collected 10 and 21 days after LPC/PBS injection and the extent of white matter damage and repair were analyzed (*Figure 1a*). Serially cut sections were stained with Fluoromyelin-Green (FM-G) and anti-GFAP (*Figure 1—figure supplement 2a and b*) or subjected to *in situ* hybridization (ISH) for the myelin-associated gene products *Mbp*, *Mag*, *Plp1* and the OPC marker *Pdgfra* (*Figure 1b* and *Figure 1—figure supplement 2c and d*). Independent of *Lrp1* genotype, at 10 days following LPC injection, similar-sized white matter lesions (area devoid of FM-G labeling) and comparable astrogliosis were observed (*Figure 1—figure supplement 2b*). At 21 DPI, however, astrogliosis was increased and the lesion area larger in LPC injected *Lrp1* iKO mice (*Figure 1—figure supplement 2b*). ISH revealed no changes in *Mbp*, *Mag*, *Plp1*, or *Pdgfra* expression on the PBS injected side (*Figure 1—figure supplement 2c*); however, LPC injection resulted in a strong increase in *Mag*, *Plp1*, and *Mbp1* (*Figure 1—figure supplement 2c and d*). Because *Mbp* mRNA is strongly upregulated in myelin producing OLs and transported into internodes (*Ainger et al., 1993*), we used *Mpb* ISH to find the white matter lesion (*Figure 1b*). The section with the largest circumference of the intensely labeled *Mbp$^+$* area was defined as the

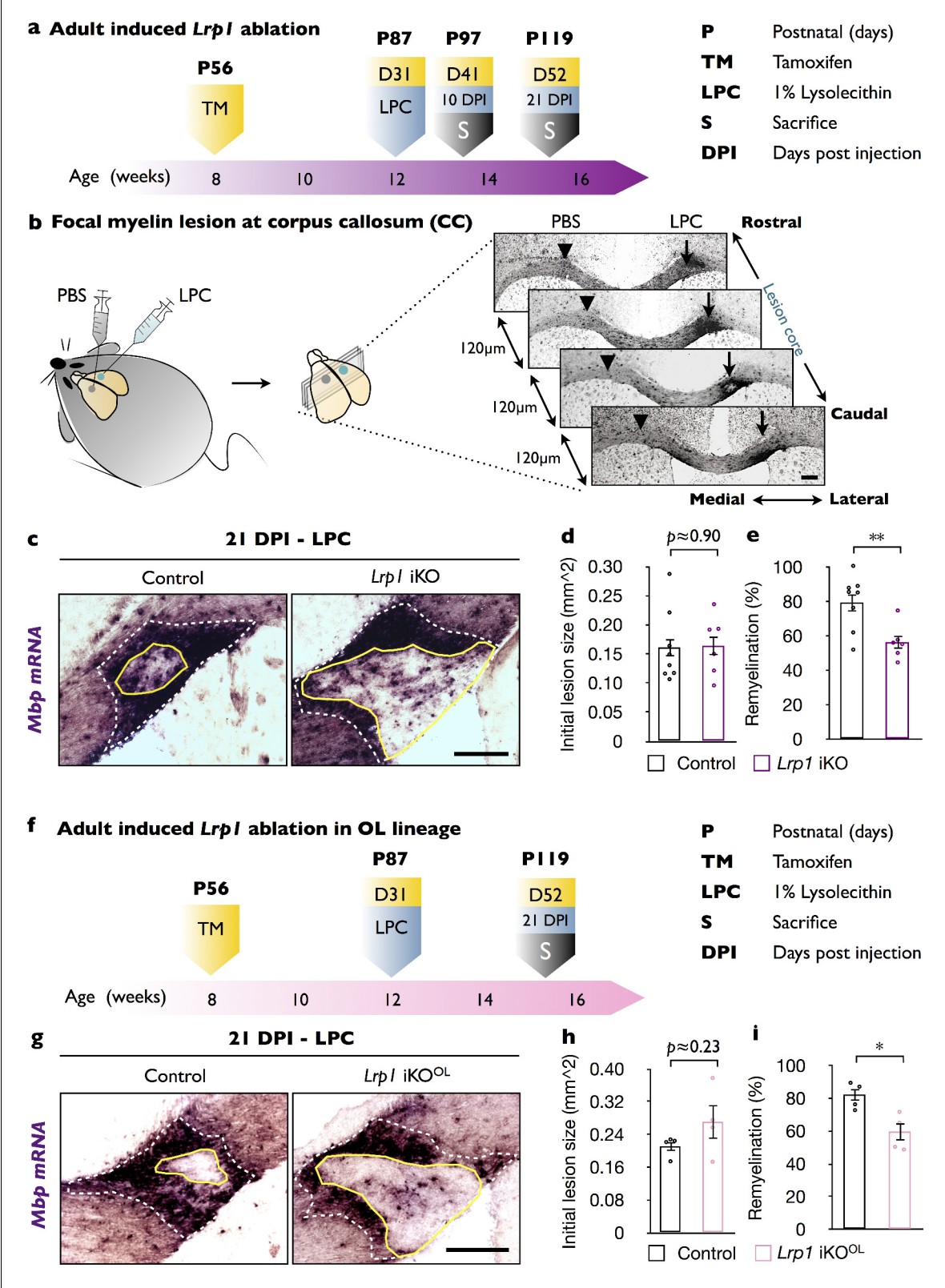

**Figure 1.** In adult mice, global and OL-lineage selective ablation of *Lrp1* attenuates white matter repair. (**a**) Timeline in weeks indicating when *Lrp1* ablation was induced (*Lrp1^flox/flox^;CAG-CreER^TM^, Lrp1* iKO), lysolecithin (LPC) injected, and animals sacrificed. (**b**) Cartoon showing unilateral injection of LPC in the corpus callosum (CC) and PBS into the contralateral side. Coronal brain sections (series of 6, each 120 μm apart) probed for *Mbp* by *in situ* hybridization (ISH). Brain sections containing the lesion center were identified and subjected to quantification. (**c**) Coronal brain sections through the

*Figure 1 continued on next page*

*Figure 1 continued*

CC 21 days post LPC injection (21 DPI). The outer rim of the lesion area (lesion^out^) is demarcated by the elevated *Mbp* signal (white dashed line). The non-myelinated area of the lesion is defined by the inner rim of elevated *Mbp* signal (lesion^in^) and delineated by a solid yellow line. Scale bar = 200 μm. (d) Quantification of the initial lesion size (lesion^out^) in *Lrp1* control (n = 8) and iKO (n = 6) mice. (e) Quantification of white matter repair in *Lrp1* control (n = 8) and iKO (n = 6) mice. The extent of repair was calculated as the percentile of (lesion^out^ - lesion^int^)/(lesion^out^) x 100. (f) Timeline in weeks showing when OL-lineage-specific *Lrp1* ablation (*Lrp1^flox/flox^;Pdgfra-CreER^TM^*, *Lrp1* iKO^OL^) was induced, LPC injected, and animals sacrificed. (g) Coronal brain sections through the CC at 21 days post LPC injection of *Lrp1* control and iKO^OL^ mice. The initial lesion area is demarcated by a white dashed-line. A solid yellow line delineates the non-myelinated area. Scale bar = 200 μm. (h) Quantification of the initial lesion size in *Lrp1* control (n = 4) and iKO^OL^ (n = 4) mice. (i) Quantification of white matter repair in *Lrp1* control (n = 4) and iKO^OL^ (n = 4) mice. Results are shown as mean ±SEM, *p<0.05, **p<0.01, and ***p<0.001, Student's *t*-test. For a detailed statistical report, see **Figure 1—source data 1**.

DOI: https://doi.org/10.7554/eLife.30498.002

The following source data and figure supplements are available for figure 1:

**Source data 1.** Raw data and detail statistical analysis report.
DOI: https://doi.org/10.7554/eLife.30498.005
**Figure supplement 1.** Generation of *Lrp1* global iKO mice.
DOI: https://doi.org/10.7554/eLife.30498.003
**Figure supplement 2.** LPC injection into the corpus callosum leads to focal white matter damage and upregulation of myelin-associated gene products.
DOI: https://doi.org/10.7554/eLife.30498.004

lesion center and subjected to quantification (**Figure 1c**). The extent of white matter lesion, the outer rim of elevated *Mbp* labeling (white dotted line), was comparable between *Lrp1* control and iKO mice (**Figure 1d**). As shown in **Figure 1c**, the area that failed to undergo repair, the inner rim of elevated *Mbp* labeling (yellow solid line), was larger in *Lrp1* iKO mice (**Figure 1c**). Quantification of lesion repair revealed a significant decrease in *Lrp1* iKO mice compared to *Lrp1* control mice (**Figure 1e**). As an independent assessment, serial sections were stained for *Pdgfra*, *Plp1*, and *Mag* transcripts and revealed fewer labeled cells within the lesion of iKO mice (**Figure 1—figure supplement 2e**). Together these studies indicate that in adult mice, *Lrp1* is required for the timely repair of a chemically induced white matter lesion. When coupled with the broad expression of *Lrp1* in different neural cell types (**Zhang et al., 2014**), this prompted further studies to examine whether *Lrp1* function in the OL lineage is important for CNS white matter repair.

## OL-lineage specific ablation of *Lrp1* impairs timely repair of damaged white matter

To determine the cell autonomy of *Lrp1* in adult white matter repair, we generated *Lrp1^flox/flox^*; *Pdgfra-CreER^TM^* (*Lrp1* iKO^OL^) mice that allow inducible gene ablation selectively in OPCs in adult mice. At P56 *Lrp1* iKO^OL^ mice were injected with TM and one month later subjected to unilateral injection of LPC into the corpus callosum and PBS on the contralateral side. *Lrp1* control mice, harboring at least one wildtype or non-recombined *Lrp1* allele, were processed in parallel. Twenty-one days post LPC/PBS injection (21 DPI), brains were collected and serially sectioned (**Figure 1f**). Detection of the initial white matter lesion and quantification of the extent of white matter repair was assessed as described above (**Figure 1b**). The initial size of the LPC inflicted white matter lesion was comparable between *Lrp1* control and iKO^OL^ mice (**Figure 1g and h**). However, the extent of lesion repair was significantly decreased in *Lrp1* iKO^OL^ mice (**Figure 1g and i**). This demonstrates an OL-lineage-specific role for *Lrp1* in the timely repair of a chemically induced white matter lesion.

## *Lrp1* is important for proper CNS myelin development and optic nerve conduction

To examine whether *Lrp1* is required for proper CNS myelin development, we generated *Lrp1^flox/flox^*;*Olig2-Cre* mice (*Lrp1* cKO^OL^) (**Figure 2—figure supplement 1a**). *Lrp1* cKO^OL^ pups are born at the expected Mendelian frequency and show no obvious abnormalities at the gross anatomical level (data not shown). LRP1 protein levels in the brains of P10, P21, and P56 *Lrp1* control and cKO^OL^ mice were analyzed by Western blot analysis and revealed a partial loss of LRP1β (**Figure 2—figure supplement 1b**). The partial loss of LRP1β in brain lysates of *Lrp1* cKO^OL^ mice is due to *Lrp1* expression in several other neural cell types. *Olig2-Cre* mice express cre recombinase under the

endogenous *Olig2* promoter, rendering mice haploinsufficient for *Olig2*. Loss of one allele of *Olig2* has been shown to reduce *Mbp* mRNA expression in neonatal mouse spinal cord (*Liu et al., 2007*). Therefore, we examined whether the presence of the *Olig2-Cre* allele influences LRP1β, MAG, CNP, or MBP in P21 brain lysates. Quantification of protein levels revealed no differences between *Lrp1*$^{flox/+}$ and *Lrp1*$^{flox/+}$;*Olig2-Cre* mice (*Figure 2—figure supplement 1c*). However, LRP1β is reduced in P10 and P21 *Lrp1*$^{flox/flox}$;*Olig2-Cre* mice, compared to *Lrp1*$^{flox/flox}$ or *Lrp1*$^{flox/+}$;*Olig2-Cre* mice (*Figure 2—figure supplement 1d*).

To examine whether *Lrp1* cKO$^{OL}$ mice exhibit defects in myelin development, optic nerves were isolated at P10, the onset of myelination; at P21, near completion of myelination; and at P56, when myelination is thought to be completed. Ultrastructural analysis at P10 revealed no significant difference in myelinated axons between *Lrp1* control (17 ± 6%) and cKO$^{OL}$(7 ± 2%) optic nerves. At P21 and P56, the percentile of myelinated axons in the optic nerve of cKO$^{OL}$ mice (49 ± 4% and 66 ± 5%, respectively) is significantly reduced compared to controls (70 ± 2% and 88 ± 1%, respectively) (*Figure 2a and b*). In *Lrp1* cKO$^{OL}$ mice, hypomyelination is preferentially observed in intermediate to small caliber axons (*Figure 2—figure supplement 1f–1n*). As an independent assessment of fiber structure, the g-ratio was determined. At P10, P21, and P56 the average g-ratio of *Lrp1* cKO$^{OL}$ optic fibers is significantly larger than in age-matched *Lrp1* control mice (*Figure 2c* and *Figure 2—figure supplement 1e*). Western blot analysis of adult *Lrp1* cKO$^{OL}$ brain lysates revealed a significant reduction in CNP, MAG, and MBP (*Figure 2—figure supplement 1o and p*). Together, these studies show that in the OL lineage *Lrp1* functions in a cell-autonomous manner and is required for proper CNS myelinogenesis.

To examine whether *Lrp1* in OLs is required for nodal organization, optic nerve sections of P21 *Lrp1* control and cKO$^{OL}$ mice were immunostained for sodium channels (PanNaCh) and the paranodal axonal protein (Caspr). Nodal density, the number of PanNaCh$^{+}$ clusters in longitudinal optic nerve sections is significantly reduced in *Lrp1* cKO$^{OL}$ mice (*Figure 2d and g*). In addition, an increase in nodal structural defects, including elongated nodes, heminodes, and nodes in which sodium channel staining is missing, was observed in mutant nerves. Quantification revealed an increase of nodal structural defects from 13.7 ± 1.3% in *Lrp1* control mice to 33.4 ± 2.9% in cKO$^{OL}$ optic nerves (*Figure 2e and h*). Sodium channel staining associated with large caliber (>1 μm) axons was increased and staining associated with small (<0.5 μm) caliber axons was reduced (*Figure 2f and i*). The density of optic nerve axons does not change between *Lrp1* control and cKO$^{OL}$ mice (*Figure 2—figure supplement 1q–1s*).

To assess whether structural defects observed in optic nerve of *Lrp1* cKO$^{OL}$ mice are associated with impaired nerve conduction, we used electrophysiological methods to measure compound action potentials (CAPs) in acutely isolated nerves (*Figure 2—figure supplement 2*). Recordings revealed a modest but significant delay in a subpopulation of myelinated optic nerve axons. The observed changes in conduction in *Lrp1* cKO$^{OL}$ nerves fit well with defects at the ultrastructural level and aberrant node assembly.

## Conditional ablation of *Lrp1* in the OL-lineage attenuates OPC differentiation

CNS hypomyelination in *Lrp1* cKO$^{OL}$ mice may be the result of reduced OPC production or impaired OPC differentiation into myelin producing OLs. To distinguish between these two possibilities, optic nerve cross-sections were stained with anti-PDGFRα, a marker for OPCs; anti-Olig2, to account for all OL lineage cells; and anti-CC1, a marker for mature OLs. No change in OPC density was observed, but the number of mature OLs was significantly reduced in *Lrp1* cKO$^{OL}$ mice (*Figure 3a and c*). Optic nerve ISH for *Pdgfra* revealed no reduction in labeled cells in *Lrp1* cKO$^{OL}$ mice, a finding consistent with anti-PDGFRα immunostaining. The density of *Plp* and *Mag* expressing cells, however, is significantly reduced in the optic nerve cross-sections and longitudinal-sections of *Lrp1* cKO$^{OL}$ mice (*Figure 3b and d*). The optic nerve cross-sectional area is not different between *Lrp1* control and cKO$^{OL}$ mice (*Figure 3—figure supplement 1a*). Together these studies show that OPCs are present at normal density and tissue distribution in *Lrp1* cKO$^{OL}$ mice, but apparently fail to generate sufficient numbers of mature, myelin-producing OLs.

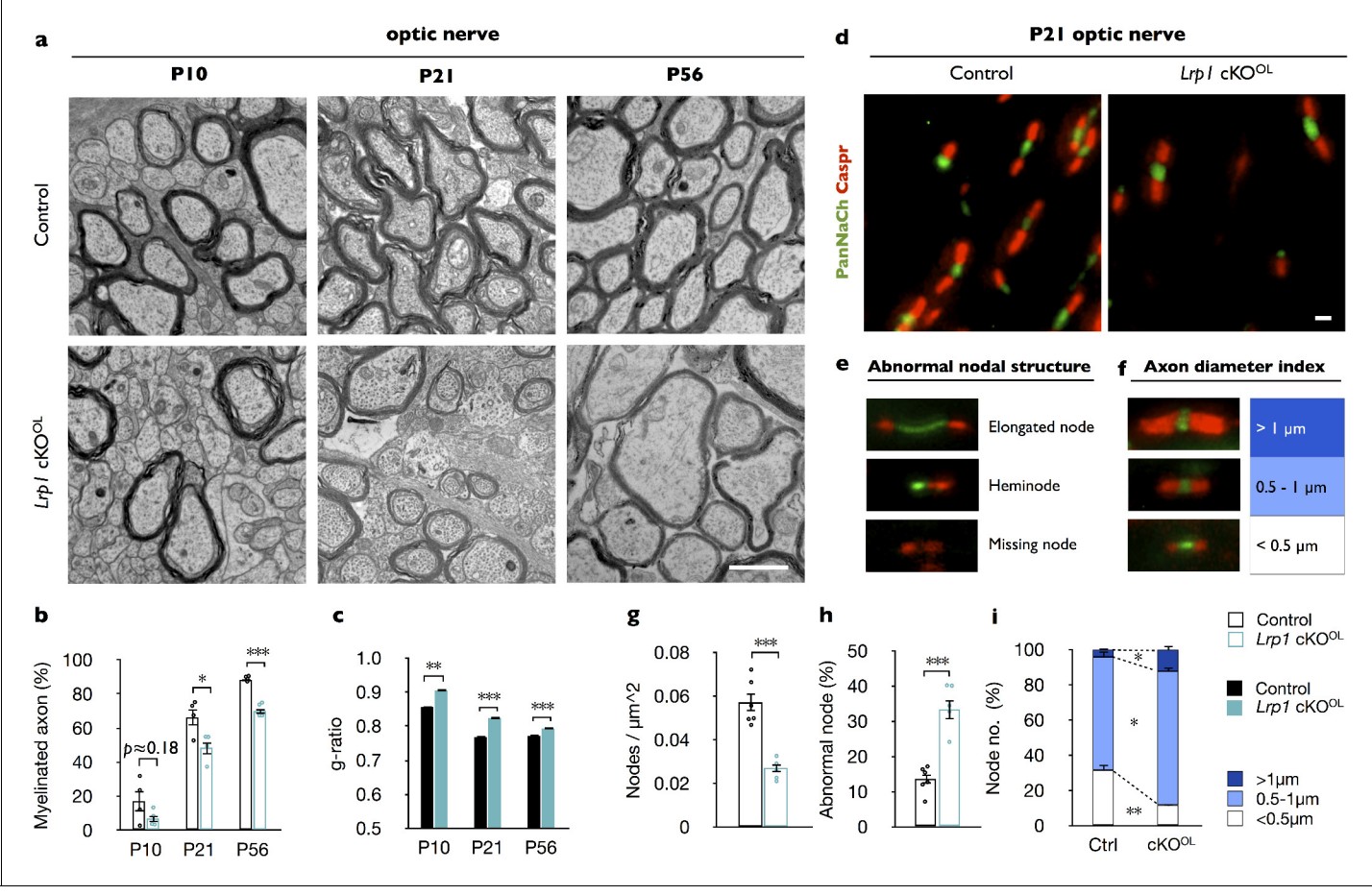

**Figure 2.** *Lrp1* ablation in the OL-lineage leads to hypomyelination and nodal defects. (a) Ultrastructural images of optic nerve cross-sections from P10, P21, and P56 control and *Lrp1*[flox/flox];*Olig2-Cre* conditional knockout mice (*Lrp1* cKO[OL]). Scale bar = 1 μm. (b) Quantification of myelinated axons in the optic nerve of *Lrp1* control and cKO[OL] mice at P10, P21 and P56 (n = 4 mice per genotype for each three time point). (c) Averaged g-ratio of *Lrp1* control and cKO[OL] optic nerve fibers from four mice per genotype for each of the three time points. At P10, n = 488 myelinated axons for control and n = 261 for cKO[OL]; at P21, n = 1015 for control and n = 997 for cKO[OL]; at P56, n = 1481 for control and n = 1020 for cKO[OL] mice. (d) Nodes of Ranvier in P21 optic nerves of *Lrp1* control and cKO[OL] mice were labeled by anti-PanNaCh (green, node) and anti-Caspr (red, paranode) staining. Scale bar = 1 μm. (e) Nodal defects detected include elongated node, heminode, and missing node ($Na^+$ channels absent). (f) Representative nodal staining categorized by axon diameter. (g) Quantification of nodal density in P21 *Lrp1* control (n = 6) and cKO[OL] (n = 5) optic nerves. (h) Quantification of abnormal nodes of Ranvier in *Lrp1* control (n = 6) and cKO[OL] (n = 5) optic nerves. (i) Quantification of nodes associated with large (>1 μm), intermediate (0.5–1 μm), and small caliber fibers (<0.5 μm) in *Lrp1* control (n = 6) and cKO[OL] (n = 5) optic nerves. Results are shown as mean ±SEM, *p<0.05, **p<0.01, and ***p<0.001, Student's *t*-test. For a detailed statistical report, see *Figure 2—source data 1*.

DOI: https://doi.org/10.7554/eLife.30498.006

The following source data and figure supplements are available for figure 2:

**Source data 1.** Raw data and detail statistical analysis report.
DOI: https://doi.org/10.7554/eLife.30498.009
**Figure supplement 1.** *Lrp1* ablation in the OL lineage leads to CNS hypomyelination.
DOI: https://doi.org/10.7554/eLife.30498.007
**Figure supplement 2.** Loss of *Lrp1* in the OL lineage leads to faulty optic nerve conduction.
DOI: https://doi.org/10.7554/eLife.30498.008

## Loss of *Lrp1* attenuates OPC differentiation *in vitro*

To independently assess the role of *Lrp1* in OL differentiation, we isolated OPCs from brains of *Lrp1* control and cKO[OL] pups (*Figure 3e*). OPCs were kept in PDGF-AA containing growth medium (GM) or switched to differentiation medium (DM) containing triiodothyronine (T3). Staining for the proliferation marker Ki67 did not reveal any change in OPC proliferation in *Lrp1* cKO[OL] cultures after 1 or 2 days in GM (*Figure 3—figure supplement 1d–1f*). After 3 days in DM, the number of NG2[+] and

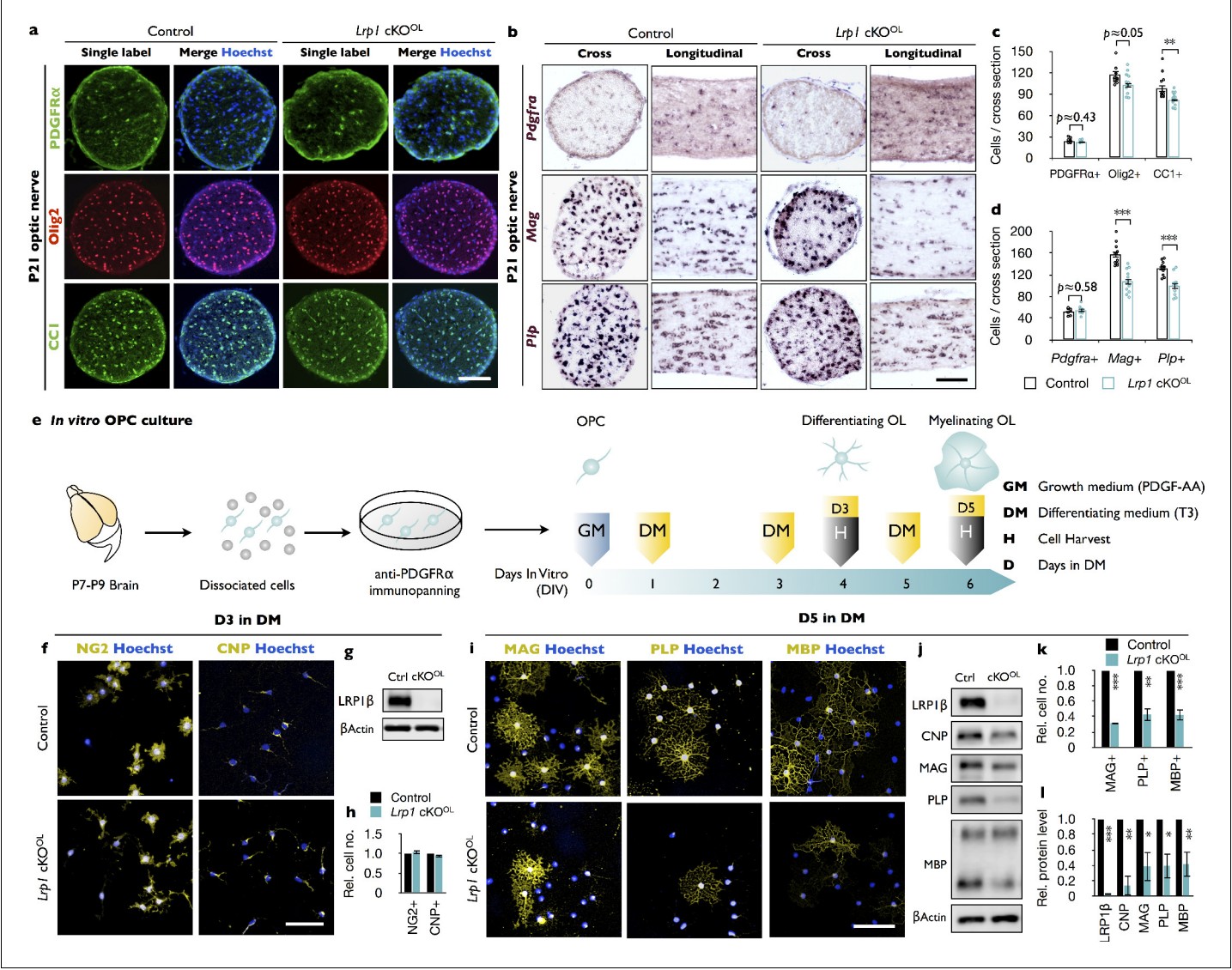

**Figure 3.** Loss of *Lrp1* in the OL-lineage attenuates OL differentiation. (**a**) Cross-sections of *Lrp1* control and cKO[OL] optic nerves stained with anti-PDGFRα (OPC marker), anti-Olig2 (pan-OL marker), anti-CC1 (mature OL marker), and Hoechst dye33342. Scale bar = 100 μm. (**b**) Cross- and -longitudinal sections of *Lrp1* control and cKO[OL] optic nerves probed for *Pdgfra*, *Mag*, and *Plp* mRNA expression. Scale bar = 100 μm. (**c**) Quantification of labeled cells per nerve cross-section. Anti-PDGFRα in control (n = 8) and cKO[OL] (n = 6) mice; anti-Olig2 and anti-CC1 in control (n = 11) and cKO[OL] (n = 12) mice. (**d**) Quantification of labeled cells per nerve cross-section. *Pdgfra*, control (n = 8) and cKO[OL] (n = 6) mice; *Mag*, control (n = 11) and cKO[OL] (n = 11) mice; *Plp*, control (n = 11) and cKO[OL] (n = 10) mice. (**e**) Workflow for OPC isolation and culturing with timeline when growth medium (GM) or differentiation medium (DM) was added and cells were harvested. (**f**) OPC/OL cultures after 3 days in DM stained with anti-NG2 (premyelinating marker), anti-CNP (differentiating OL marker), and Hoechst dye33342. Scale bar = 100 μm. (**g**) Immunoblot of OL lysates prepared from *Lrp1* control and cKO[OL] cultures after 3 days in DM probed with anti-LRP1β and anti-β-actin. (**h**) Quantification of NG2[+] (n = 3) and CNP[+] (n = 3) cells in *Lrp1* control and cKO[OL] cultures. (**i**) Control and *Lrp1* deficient OL cultures after 5 days in DM stained with anti-MAG, anti-PLP, and anti-MBP. Scale bar = 100 μm. (**j**) Immunoblotting of OL lysates prepared from *Lrp1* control and cKO[OL] cultures after 5 days in DM probed with anti-LRP1β, anti-CNP, anti-MAG, anti-PLP, anti-MBP, and anti-β-actin. (**k**) Quantification of MAG[+], PLP[+], and MBP[+] cells in *Lrp1* control (n = 3) and cKO[OL] (n = 3) cultures. (**l**) Quantification of protein levels in OL lysates detected by immunoblotting. Anti-LRP1, CNP, and PLP, n = 3 per condition; anti-MAG, n = 4 per condition; anti-MBP n = 5 per condition. Results are shown as mean values ± SEM, *p<0.05, **p<0.01, and ***p<0.001, Student's *t*-test. For a detailed statistical report, see *Figure 3—source data 1*.

DOI: https://doi.org/10.7554/eLife.30498.010

The following source data and figure supplement are available for figure 3:

**Source data 1.** Raw data and detail statistical analysis report.

DOI: https://doi.org/10.7554/eLife.30498.012

**Figure supplement 1.** Loss of *Lrp1* does not alter optic nerve size and OPC proliferation.

*Figure 3 continued on next page*

*Figure 3 continued*

DOI: https://doi.org/10.7554/eLife.30498.011

CNP[+] OLs was comparable between *Lrp1* control and cKO[OL] cultures (*Figure 3f and h*). An abundant signal for LRP1β was detected in *Lrp1* control lysate, but LRP1 was not detectable in *Lrp1* cKO[OL] cell lysate, demonstrating efficient gene deletion in the OL linage (*Figure 3g*). Moreover, a significant reduction in CNP, MAG, and PLP was detected in *Lrp1* cKO[OL] cell lysates (*Figure 3—figure supplement 1g and i*). Importantly, halplosufficiency for Olig2 in cultures prepared from *Lrp1[flox/+]* and *Lrp1[flox/+];Olig2-Cre* pups, did not reveal any difference in MAG, PLP, or LRP1β protein (*Figure 3—figure supplement 1b and c*). As LRP1 signaling is known to regulate ERK1/2 and AKT activity (*Yoon et al., 2013*), immunoblots were probed for pAKT(S473) and pErk1/2. When normalized to total AKT, levels of pAKT are reduced in *Lrp1* cKO[OL] lysate, while pERK1/2 levels are comparable between *Lrp1* control and cKO[OL] lysates (*Figure 3—figure supplement 1h and j*). Extended culture of *Lrp1*-deficient OLs in DM for 5 days is not sufficient to restore myelin protein levels. Compared to *Lrp1* control cultures, mutants show significantly fewer MAG[+], PLP[+], and MBP[+] cells (*Figure 3i and k*) and immunoblotting of cell lysates revealed a reduction in total CNP, MAG, PLP, and MBP (*Figure 3j and l*). Collectively, our studies demonstrate a cell-autonomous function for *Lrp1* in the OL lineage, important for OPC differentiation into myelin sheet producing OLs.

## *Lrp1* deficiency in OPCs and OLs causes a reduction in free cholesterol

While LRP1 has been implicated in cholesterol uptake and homeostasis in non-neural cell types (*van de Sluis et al., 2017*), a role in cholesterol homeostasis in the OL-lineage has not yet been investigated. We find that *Lrp1[−/−]* OPCs, prepared from *Lrp1[flox/flox];Olig2-Cre*, have reduced levels of free cholesterol compared to *Lrp1* control OPCs (*Figure 4a and b*). Levels of cholesteryl-ester are very low in the CNS (*Björkhem and Meaney, 2004*) and near the detection limit in *Lrp1* control and *Lrp1[−/−]* OPCs (*Figure 4c*). Morphological studies with MBP[+] OLs revealed a significant reduction in myelin-like membrane sheet expansion in *Lrp1[−/−]* OLs (*Figure 4d and e*), reminiscent of wildtype OLs cultures treated with statins to inhibit HMG-CoA reductase, the rate limiting enzyme in the cholesterol biosynthetic pathway (*Maier et al., 2009*; *Paintlia et al., 2010*; *Smolders et al., 2010*). To assess cholesterol distribution in primary OLs, cultures were stained with filipin. In *Lrp1* control OLs, staining was observed on myelin sheets and was particularly strong near the cell soma. In *Lrp1[−/−]* OLs, filipin and MBP staining were significantly reduced (*Figure 4f*). Reduced filipin staining is not simply a reflection of smaller cell size, as staining intensity was decreased when normalized to myelin sheet surface area (*Figure 4g*). Thus, independent measurements revealed a dysregulation of cholesterol homeostasis in *Lrp1[−/−]* OPCs/OLs.

Cellular lipid homeostasis is regulated by a family of membrane-bound basic helix-loop-helix transcription factors, called sterol-regulatory element-binding proteins (SREBPs). To assess whether *Lrp1* deficiency leads to an increase in SREBP2, OLs were cultured for 3 days in DM and analyzed by immunoblotting. OL cultures prepared from *Lrp1[flox/+]* and *Lrp1[flox/+];Olig2-cre* pups showed very similar levels of SREBP2. In marked contrast, we observed a strong upregulation of SREBP2 in *Lrp1[−/−]* cultures (*Figure 4i* and *Figure 4—figure supplement 1a and b*). Elevated SREBP2 in mutant cultures can be reversed by exogenous cholesterol directly added to the culture medium (*Figure 4i and j*). This shows the existence of LRP1-independent cholesterol uptake mechanisms in *Lrp1[−/−]* OLs and a normal physiological response to elevated levels of cellular cholesterol. In *Lrp1* control cultures, bath application of cholesterol leads to a small, yet significant decrease in SREBP2 (*Figure 4j*). Given the importance of cholesterol in OL maturation (*Krämer-Albers et al., 2006*; *Mathews et al., 2014*; *Saher et al., 2005*), we examined whether the differentiation block can be rescued by bath-applied cholesterol. Remarkably, cholesterol treatment of *Lrp1[−/−]* OLs for 3 days failed to augment PLP, MAG or CNP to control levels (*Figure 4k–n*). While cholesterol treated *Lrp1[−/−]* OLs showed a modest increase in PLP, levels remained below *Lrp1* controls. Moreover, prolonged cholesterol treatment for 5 days failed to increase PLP levels (*Figure 4o and p*) or the number of MBP[+] OLs in *Lrp1[−/−]* cultures (*Figure 4q and r*). Although differentiation of *Lrp1[−/−]* OLs cannot be 'rescued' by bath applied cholesterol, cells are highly sensitive to a further reduction in cholesterol, as shown by bath applied simvastatin (*Figure 4—figure supplement 1e and f*). Since cholesterol is

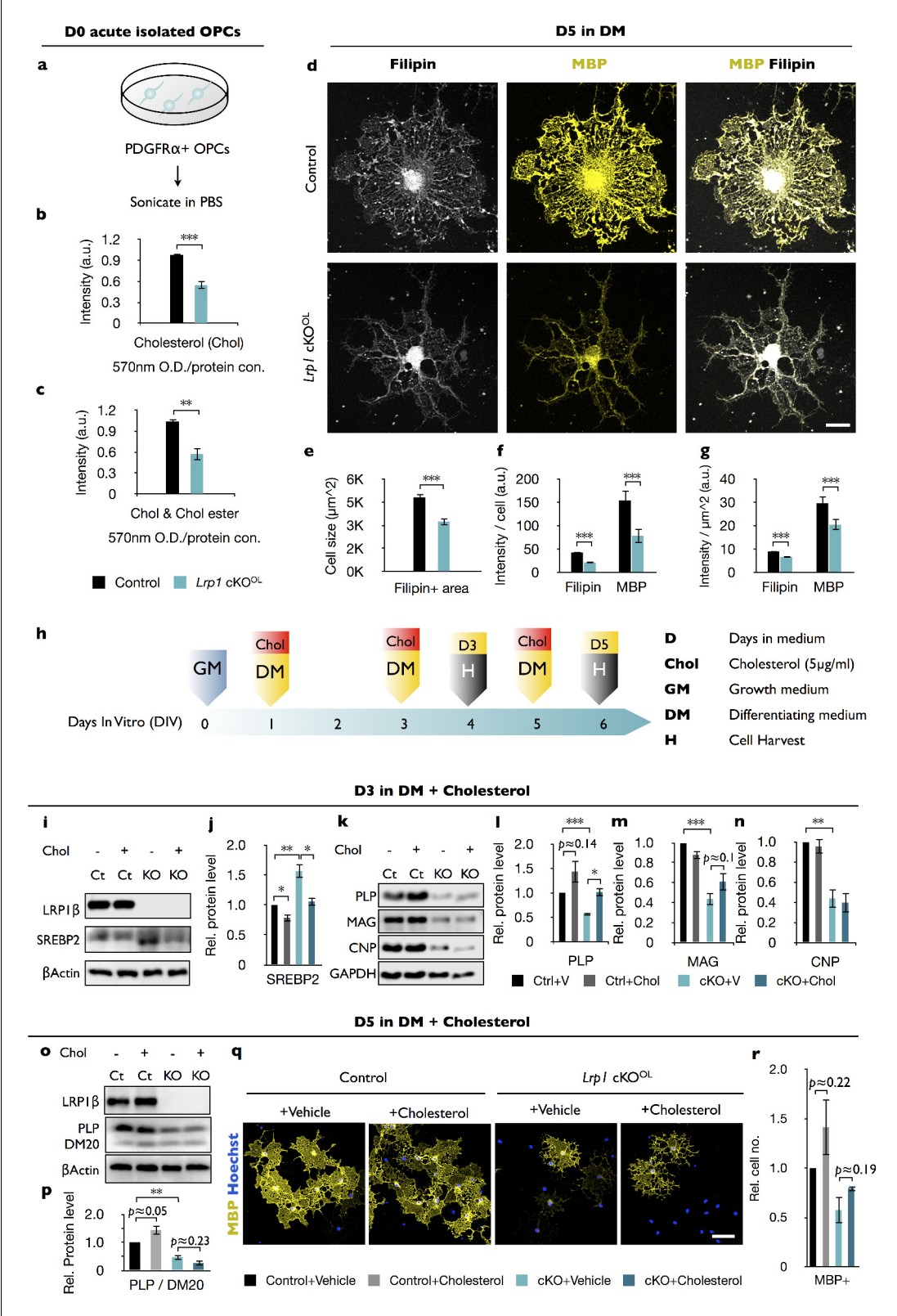

**Figure 4.** Free cholesterol is reduced in OPCs deficient for *Lrp1*. (a) OPCs were isolated from P8 brains by anti-PDGFRα immunopanning, sonicated and subjected to measurement of cholesterol (Chol). (b and c) Quantification of free Chol (b) and total Chol (Chol and Chol ester) (c) in OPCs isolated from *Lrp1* control (n = 5) and cKO[OL] (n = 5) mouse pups. (d) *Lrp1* control and cKO[OL] OLs after 5 days in DM stained with filipin and anti-MBP. Scale bar = 10 μm. (e–g) Quantification of OL size in μm[2] (e), the intensity of filipin and MBP labeling per cell (f), and the intensity of filipin and MBP staining

*Figure 4 continued on next page*

*Figure 4 continued*

per µm$^2$ (g). For *Lrp1* control and cKO$^{OL}$ OLs, n = 29 cells from three mice in each group. (h) Timeline in days showing when growth medium (GM) or differentiation medium (DM) with (+) or without (-) Chol was added and when cells were harvested. (i and k) Immunoblotting of OL lysates prepared from *Lrp1* control and cKO$^{OL}$ cultures after 3 days in DM. Representative blots were probed with anti-LRP1β, anti-SREBP2, anti-β-actin, anti-PLP, anti-MAG, anti-CNP, and anti-GAPDH. (j, l–n) Quantification of SREBP2 (j), PLP (l), MAG (m), and CNP (n) in *Lrp1* control and cKO$^{OL}$ cultures ± bath applied Chol. Number of independent immunoblots: anti-PLP and MAG, n = 3 per condition; anti-SREBP2 and anti-CNP, n = 4 per condition. (o) Immunoblotting of OL lysates prepared from *Lrp1* control and cKO$^{OL}$ cultures after 5 days in DM ±bath applied Chol. Representative blots were probed with anti-LRP1β, anti-PLP/DM20, and anti-β-actin. (p) Quantification of PLP (n = 4 per condition) in *Lrp1* control and cKO$^{OL}$ cultures ± bath applied Chol (q) Immunostaining of OLs after 5 days in DM ±bath applied Chol. Primary OLs stained with anti-MBP and Hoechst dye33342. Scale bar = 100 µm. (r) Quantification showing relative number of MBP$^+$ cells in *Lrp1* control and cKO$^{OL}$ cultures (n = 3–5 per condition). Results are shown as mean values ± SEM, *p<0.05, **p<0.01, and ***p<0.001, 2-way ANOVA, post hoc *t*-test. For a detailed statistical report, see *Figure 4—source data 1*.
DOI: https://doi.org/10.7554/eLife.30498.013

The following source data and figure supplement are available for figure 4:

**Source data 1.** Raw data and detail statistical analysis report.
DOI: https://doi.org/10.7554/eLife.30498.015
**Figure supplement 1.** *Lrp1*-deficient OLs are sensitive to statin treatment but not to bath applied mevalonate.
DOI: https://doi.org/10.7554/eLife.30498.014

only one of many lipid derivatives produced by the cholesterol biosynthetic pathway (*Figure 4—figure supplement 1c*), we asked whether treatment with mevalonate improves differentiation of *Lrp1*$^{-/-}$ OPC. However, similar to cholesterol, mevalonate fails to increase differentiation (*Figure 4—figure supplement 1g and h*). Taken together, *Lrp1* deficiency in the OL-lineage leads to a drop in cellular cholesterol and arrest in differentiation that cannot be rescued by cholesterol or mevalonate supplementation. Our data suggest that in addition to cholesterol homeostasis, LRP1 regulates other biological processes important for OPC differentiation.

### *Lrp1* deficiency impairs peroxisome biogenesis

To further investigate what type of biological processes might be dysregulated by *Lrp1* deficiency, we performed transcriptomic analyses of OPCs acutely isolated from *Lrp1* control and cKO$^{OL}$ pups. Gene ontology (GO) analysis identified differences in 'peroxisome organization' and 'peroxisome proliferation-associated receptor (PPAR) signaling pathway' (*Figure 5—figure supplement 1a*). Six gene products regulated by *Lrp1* belong to peroxisome and PPAR GO terms, including *Pex2*, *Pex5l*, *Hrasls*, *Ptgis*, *Mavs*, and *Stard10* (*Figure 5—figure supplement 1b*). Western blot analysis of *Lrp1*$^{-/-}$ OLs further revealed a significant reduction in PEX2 after 5 days in DM (*Figure 5—figure supplement 1c and d*). Because PEX2 has been implicated in peroxisome biogenesis (*Gootjes et al., 2004*), and peroxisome biogenesis disorders (PBDs) are typically associated with impaired lipid metabolism and CNS myelin defects (*Krause et al., 2006*), this prompted us to further explore a potential link between LRP1 and peroxisomes. To assess whether the observed reduction in PEX2 impacts peroxisome density in primary OLs, MBP$^+$ OLs were stained with anti-PMP70, an ATP-binding cassette transporter enriched in peroxisomes (*Figure 5a*). In *Lrp1*$^{-/-}$ OLs, we observed reduced PMP70 staining (*Figure 5b*) and a decrease in the total number of peroxisomes (*Figure 5c*). Normalization of peroxisome counts to cell size revealed that the reduction in *Lrp1*$^{-/-}$ OLs is not simply a reflection of smaller cells (*Figure 5d*). The subcellular localization of peroxisomes is thought to be important for ensuring a timely response to metabolic demands (*Berger et al., 2016*). This prompted us to analyze the distribution of peroxisomes in primary OLs. Interestingly, while the number of PMP70 positive puncta near the cell soma is comparable between *Lrp1* control and *Lrp1*$^{-/-}$ OLs, we observed a significant drop in peroxisomes along radial processes of MBP$^+$ OLs (*Figure 5e–h*).

### Combination treatment of cholesterol and PPARγ agonist rescues the differentiation block in *Lrp1*-deficient OPCs

In endothelial cells, the LRP1-ICD functions as a co-activator of PPARγ, a key regulator of lipid and glucose metabolism (*Mao et al., 2017*). Activated PPARγ moves into the nucleus to control gene expression by binding to PPAR-responsive elements (PPREs) on numerous target genes, including *Lrp1* (*Gauthier et al., 2003*). In addition, PPREs are found in genes important for lipid and glucose

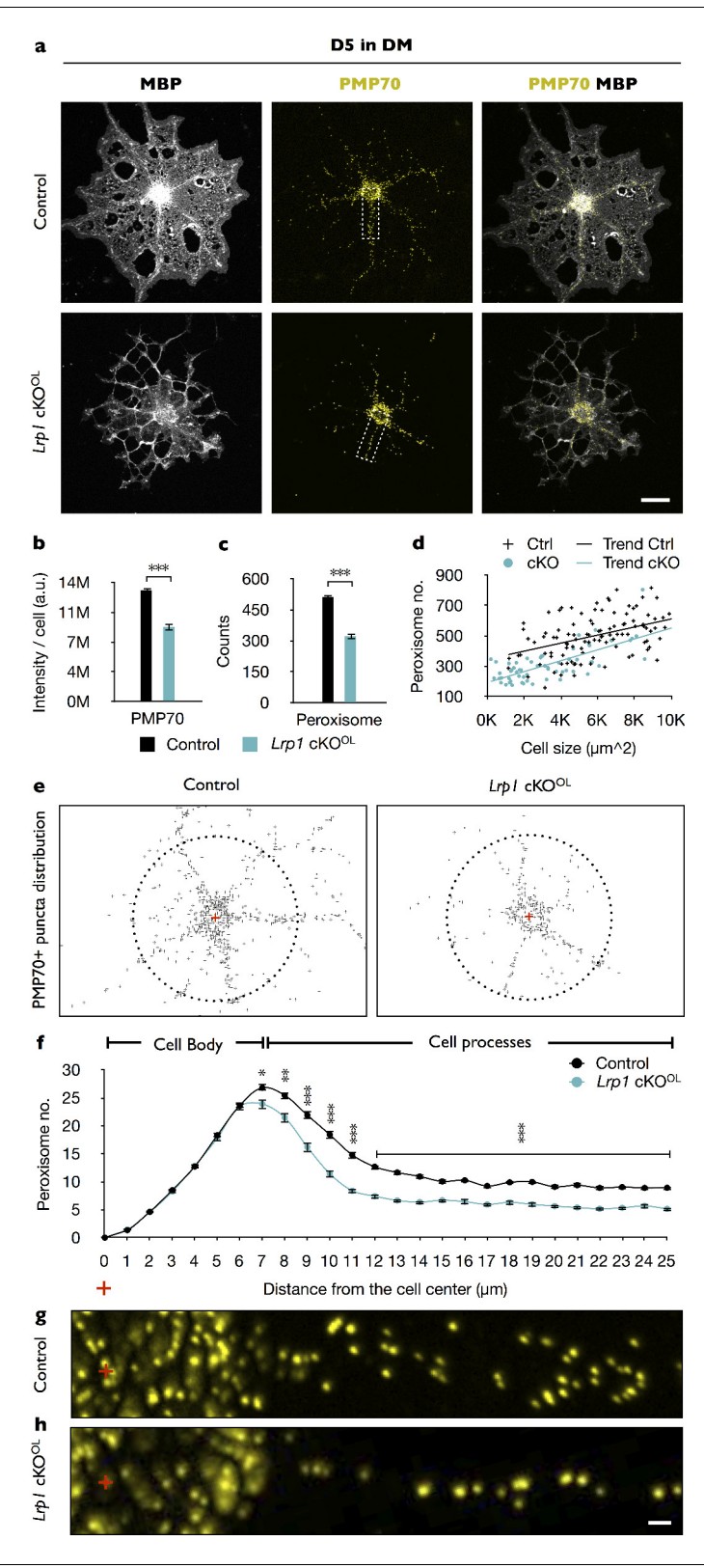

**Figure 5.** In primary OLs, peroxisome density and distribution is regulated by *Lrp1*. (a) Primary OLs prepared from *Lrp1* control and cKO[OL] OL pups, cultured for 5 days in DM were stained with anti-MBP and anti-PMP70. Scale bar = 10 μm. (b–d) Quantification of PMP70 labeling intensity per cell (b), PMP70[+] puncta per cell (c), and scatter plot showing the number of PMP70[+] peroxisomes as a function of cell size for MBP[+] OLs of *Lrp1* control and *Lrp1* cKO[OL] cultures (d). For *Lrp1* control OLs, n = 112 cells from three mice. For *Lrp1* cKO[OL] OLs, n = 60 cells from three mice. (e) Representative

*Figure 5 continued on next page*

*Figure 5 continued*

distribution of PMP70$^+$ puncta of *Lrp1* control and cKO$^{OL}$ OL. For quantification, the center of the cell was marked with a red cross. Puncta within a 25 µm radius from the center (dashed circle) were subjected to quantification. (f) Quantification of peroxisome number plotted against the distance from the center of *Lrp1* control (n = 113 cells, three mice) and cKO$^{OL}$ (n = 63 cells, three mice) OLs. (g and h) Representative high-magnification views of PMP70$^+$ puncta from areas boxed in panel (a). Scale bar = 1 µm. Results are shown as mean values ± SEM, *p<0.05, **p<0.01, and ***p<0.001, Student's *t*-test. For a detailed statistical report, see *Figure 5—source data 1*.

DOI: https://doi.org/10.7554/eLife.30498.016

The following source data and figure supplement are available for figure 5:

**Source data 1.** Raw data and detail statistical analysis report.
DOI: https://doi.org/10.7554/eLife.30498.018

**Figure supplement 1.** Gene ontology (GO) analysis of *Lrp1*-deficient OPCs revealed enrichment of peroxisomal genes.
DOI: https://doi.org/10.7554/eLife.30498.017

metabolism, and peroxisome biogenesis (*Fang et al., 2016*; *Hofer et al., 2017*). *In vitro*, a 5-day treatment of *Lrp1* control OPCs with pioglitazone, an agonist of PPARγ, results in elevated LRP1 (*Figure 6a and b*) and accelerated differentiation into MBP$^+$ OLs (*Figure 6c and d*) (*Bernardo et al., 2009*). This stands in marked contrast to *Lrp1$^{-/-}$* cultures, where pioglitazone treatment fails to accelerate OPC differentiation (*Figure 6c and d*). Moreover, pioglitazone does not regulate PMP70 staining intensity in MBP$^+$ *Lrp1* control or *Lrp1$^{-/-}$* OLs, nor does it have any effect on total peroxisome counts per cell (*Figure 6e–i*). However, pioglitazone leads to a modest but significant increase in the number of peroxisomes located in cellular processes of *Lrp1$^{-/-}$* OLs (*Figure 6j and k*). Treatment of *Lrp1* control OPCs with the PPARγ antagonist GW9662 blocks differentiation into MBP$^+$ OLs (*Roth et al., 2003*), but does not lead to a further reduction in MBP$^+$ cells in *Lrp1$^{-/-}$* OL cultures (*Figure 6l and m*). This suggests that in *Lrp1$^{-/-}$* OLs PPARγ is not active.

Given LRP1's multifunctional receptor role, we asked whether simultaneous treatment with pioglitazone and cholesterol is sufficient to rescue the differentiation block of *Lrp1$^{-/-}$* OPCs (*Figure 7a*). This is indeed the case, as the number of MBP$^+$ cells in *Lrp1$^{-/-}$* cultures is significantly increased by the combination treatment (*Figure 7b and c*). Moreover, the size of MBP$^+$ *Lrp1$^{-/-}$* OLs increased (*Figure 7d and f*) and peroxisome counts are elevated (*Figure 7e and g*), however the anti-MBP staining intensity was only partially rescued (*Figure 7h*). Quantification of peroxisome distribution in *Lrp1$^{-/-}$* OPC/OL cultures subjected to combo treatment revealed a marked increase in PMP70$^+$ peroxisomes in OL processes (*Figure 7i and j*). Together, these findings indicate that LRP1 regulates multiple metabolic functions important for OL differentiation. In addition to its known role in cholesterol homeostasis, LRP1 regulates expression of PEX2 and thereby metabolic functions associated with peroxisomes.

## Discussion

LRP1 function in the OL-lineage is necessary for proper CNS myelin development and the timely repair of a chemically induced focal white matter lesion *in vivo*. Optic nerves of *Lrp1* cKO$^{OL}$ show fewer myelinated axons, thinning of myelin sheaths, and an increase in nodal structural defects. Morphological alterations have a physiological correlate, as *Lrp1* cKO$^{OL}$ mice exhibit faulty nerve conduction. Mechanistically, *Lrp1* deficiency disrupts multiple signaling pathways implicated in OL differentiation, including AKT activation, cholesterol homeostasis, PPARγ signaling, peroxisome biogenesis and subcellular distribution. The pleiotropic roles of LRP1 in OPC differentiation are further underscored by the fact that restoring cholesterol homeostasis or activation of PPARγ alone is not sufficient to drive differentiation. Only when cholesterol supplementation is combined with PPARγ activation, is differentiation of *Lrp1$^{-/-}$* OPC into MBP$^+$ OLs significantly increased. Taken together, our studies identify a novel role for LRP1 in peroxisome function and suggest that broad metabolic dysregulation in *Lrp1*-deficient OPCs attenuates differentiation into mature OLs (*Figure 8*).

In the embryonic neocortex, LRP1 is strongly expressed in the ventricular zone and partially overlaps with nestin$^+$ neural stem and precursor cells (*Hennen et al., 2013*). In *Lrp1$^{flox/flox}$* neurospheres, conditional gene ablation reduces cell proliferation, survival, and negatively impacts differentiation into neurons and O4$^+$ OLs (*Safina et al., 2016*). Consistent with these observations, *Lrp1* cKO$^{OL}$ mice show reduced OPC differentiation *in vivo*. Studies with purified OPCs *in vitro* and OL-linage

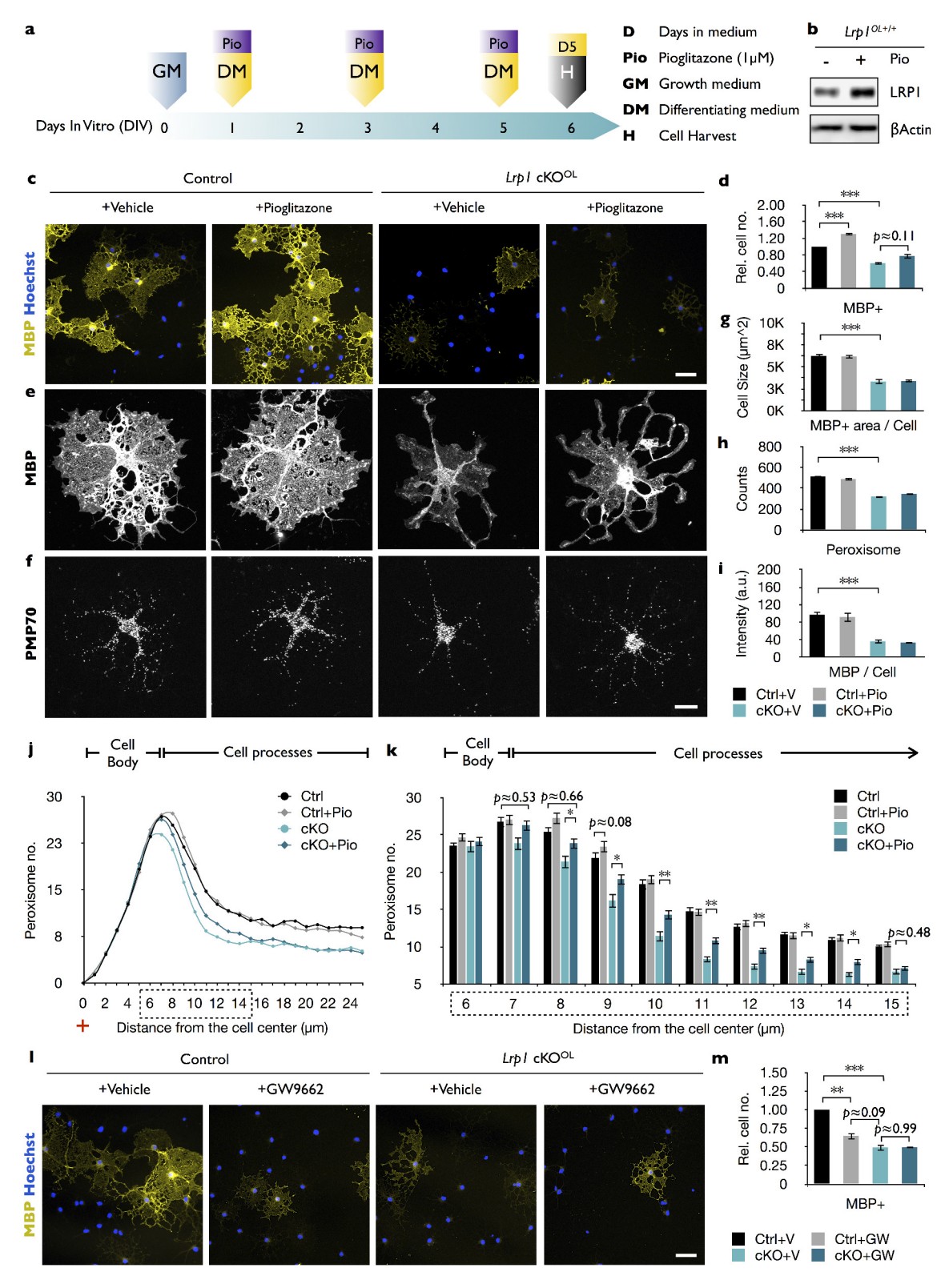

**Figure 6.** In *Lrp1*-deficient OPCs, PPARγ activation increases peroxisome density but does not promote cell differentiation. (**a**) Timeline in days showing when growth medium (GM) or differentiation medium (DM) with pioglitazone (Pio) were supplied and cells were harvested for analysis. (**b**) Immunoblots of OL lysates prepared from *Lrp1* wildtype cultures after 5 days in DM with (+) or without (-) Pio, probed with anti-LRP1β. Anti-β-actin is shown as loading control. (**c**) Immunostaining of *Lrp1* control and cKO[OL] cultures after 5 days in DM. Representative cell cultures stained with anti-MBP and

*Figure 6 continued on next page*

Figure 6 continued

Hoechst dye 33342. Scale bar = 50 μm. (d) Quantification of MBP$^+$ cells in *Lrp1* control cultures with vehicle (n = 6), *Lrp1* control cultures with Pio (n = 6), cKO$^{OL}$ cultures with vehicle (n = 4), and cKO$^{OL}$ cultures with Pio (n = 4). (e–f) Primary OLs probed with anti-MBP and anti-PMP70. Scale bar = 10 μm. (g–i) Quantification of OL size in μm$^2$ (g), the number of PMP70$^+$ puncta (h), and the intensity of MBP staining per cell (i). (j) Distribution of peroxisomes as a function of distance from the cell center in *Lrp1* control and cKO$^{OL}$ OLs treated ±Pio. The number of PMP70$^+$ peroxisomes between 6–15 μm in *Lrp1* control and cKO$^{OL}$ cultures was subjected to statistical analysis in (k). *Lrp1* control (n = 112 cells, three mice), *Lrp1* control cultures with Pio (n = 180 cells, three mice), cKO$^{OL}$ (n = 60 cells, three mice), and cKO$^{OL}$ cultures with Pio. (n = 110 cells, three mice) (k). (l) Immunostaining of OLs after 5 days in DM ±GW9662, probed with anti-MBP and Hoechst dye33342. Scale bar = 50 μm. (m) Quantification of MBP$^+$ cells under each of the four different conditions (n = 3 per condition). Results are shown as mean values ± SEM, *p<0.05, **p<0.01, and ***p<0.001, 2-way ANOVA, post hoc *t*-test. For a detailed statistical report, see *Figure 6—source data 1*.
DOI: https://doi.org/10.7554/eLife.30498.019

The following source data is available for figure 6:

**Source data 1.** Raw data and detail statistical analysis report.
DOI: https://doi.org/10.7554/eLife.30498.020

specific gene ablation *in vivo*, suggest a cell-autonomous role for *Lrp1* in OPC maturation. Non-cell-autonomous functions for LRP1 following white matter lesion are likely, since LRP1 is upregulated in astrocytes and myeloid cells near multiple sclerosis lesions (*Chuang et al., 2016*). Moreover, deletion of *Lrp1* in microglia worsens the course of experimental autoimmune encephalomyelitis and has been proposed to promote a proinflammatory milieu associated with disease exacerbation (*Chuang et al., 2016*). Studies with *Lrp1 iKO$^{OL}$* mice show that *Lrp1* in the OL linage is necessary for the timely repair of a focal myelin lesion. This suggests that similar to OPCs in the developing brain, OPCs in the adult brain depend on *Lrp1* for rapid differentiation into myelin producing OLs. Since white matter repair was analyzed by repopulation of the lesion area with *Mbp$^+$* cells, additional studies, including electron microscopy, will be needed to demonstrate a requirement for *Lrp1* in remyelination of denuded axons.

Cholesterol does not cross the blood-brain-barrier (*Saher and Stumpf, 2015*) and CNS resident cells need to either synthesize their own cholesterol or acquire it through horizontal transfer from neighboring cells, including astrocytes (*Camargo et al., 2017*). In the OL-lineage cholesterol is essential for cell maturation, including myelin gene expression, myelin protein trafficking, and internode formation (*Krämer-Albers et al., 2006*; *Mathews et al., 2014*; *Saher et al., 2005*). Sterol biosynthesis is in part accomplished by peroxisomes. Specifically, the pre-squalene segment of the cholesterol biosynthetic pathway takes place in peroxisomes. However, cholesterol is only one of many lipid derivatives produced by this pathway (*Faust and Kovacs, 2014*). A drop in intracellular cholesterol leads to an increase in SREBPs, a family of transcription factors that regulate expression of gene products involved in cholesterol and fatty acid synthesis (*Faust and Kovacs, 2014*; *Goldstein et al., 2006*). In Schwann cells, SREBPs and the SREBP-activating protein SCAP are required for AKT/mTOR-dependent lipid biosynthesis, myelin membrane synthesis, and normal PNS myelination (*Norrmén et al., 2014*; *Verheijen et al., 2009*). In the OL linage blockage of SREBP inhibits CNS myelination (*Camargo et al., 2017*; *Monnerie et al., 2017*). Blocking of SREBP processing in primary OLs leads to a drop in cholesterol and inhibits cell differentiation and membrane expansion. This can be rescued by cholesterol added to the culture medium (*Monnerie et al., 2017*). In primary OLs, *Lrp1* deficiency leads to activation of SREBP2, yet cells are unable to maintain cholesterol homeostasis, suggesting more global metabolic deficits. The cholesterol sensing apparatus in *Lrp1*-deficient OPCs appears to be largely intact, as bath applied cholesterol restores SREBP2 to control levels. Since SREBP2 can be induced by ER stress (*Faust and Kovacs, 2014*), reversibility by bath applied cholesterol suggests that *Lrp1* cKO$^{OL}$ cultures upregulate SREBP2 due to cholesterol deficiency and not elevated ER stress (*Faust and Kovacs, 2014*). Significantly, restoring cellular cholesterol homeostasis in *Lrp1$^{-/-}$* OPCs is not sufficient to overcome the differentiation block, suggesting more widespread functional deficits.

Members of the PPAR subfamily, including PPARα, PPARβ/δ, and PPARγ, are ligand-activated transcription factors that belong to the nuclear hormone receptor family (*Michalik et al., 2006*). PPARs regulate transcription through heterodimerization with the retinoid X receptor (RXR). When activated by a ligand, the dimer modulates transcription via binding to a PPRE motif in the promoter region of target genes (*Michalik et al., 2006*). PPARs-regulated gene expression controls numerous

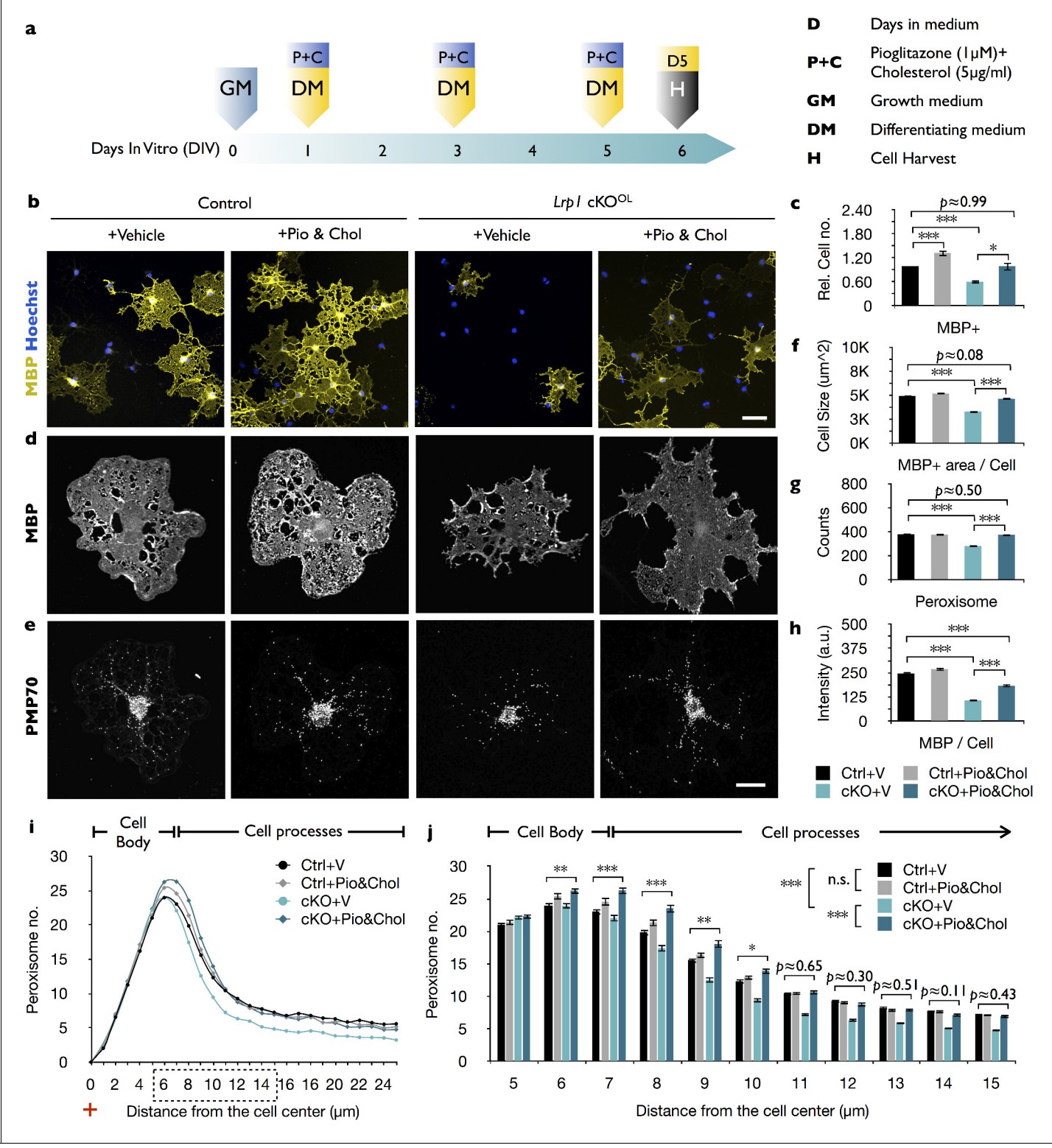

**Figure 7.** The combined treatment with cholesterol and pioglitazone rescues the differentiation block of *Lrp1* deficient OPCs. (**a**) Timeline in days showing when growth medium (GM) or differentiation medium (DM) with pioglitazone (Pio) and cholesterol (Chol) was supplied and cells were harvested for analysis. (**b**) Immunostaining of *Lrp1* control and cKO[OL] cultures after 5 days in DM, probed with anti-MBP and Hoechst dye 33342. Scale bar = 50 µm. (**c**) Quantification of MBP[+] cells in *Lrp1* control cultures treated with vehicle (n = 4), *Lrp1* control cultures treated with Pio and Chol (n = 3), cKO[OL] cultures treated with vehicle (n = 4), and cKO[OL] cultures treated with Pio and Chol (n = 3). (**d and e**) Primary OLs probed with anti-MBP and anti-PMP70. Scale bar = 10 µm. (**f–h**) Quantification of OL size in µm[2] (**f**), the number of PMP70[+] puncta and (**g**), the intensity of MBP staining per cell (**h**). (**i**)

*Figure 7 continued on next page*

*Figure 7 continued*

Distribution of peroxisomes as a function of distance from the cell center in *Lrp1* control and cKO[OL] cultures with (+) or without (-) Pio and Chol combotreatment. The number of PMP70[+] peroxisomes between 5–15 μm in *Lrp1* control and cKO[OL] cultures was subjected to statistical analysis in (j). *Lrp1* control cultures with vehicle (n = 210 cells, three mice), *Lrp1* control cultures treated with Pio and Chol (n = 208 cells, three mice), cKO[OL] cultures treated wtih vehicle (n = 199 cells, three mice), and cKO[OL] cultures treated wtih Pio and Chol (n = 190 cells, three mice) (k). Results are shown as mean values ± SEM, *p<0.05, **p<0.01, and ***p<0.001, 2-way ANOVA, post hoc *t*-test. For a detailed statistical report, see *Figure 7—source data 1*.

DOI: https://doi.org/10.7554/eLife.30498.021

The following source data is available for figure 7:

**Source data 1.** Raw data and detail statistical analysis report.

DOI: https://doi.org/10.7554/eLife.30498.022

biochemical pathways implicated in lipid, glucose and energy metabolism (*Berger and Moller, 2002*; *Han et al., 2017*). A critical role for PPARγ in OL differentiation is supported by the observation that activation with pioglitazone or rosiglitazone accelerates OPC differentiation into mature OLs (*Bernardo et al., 2009*; *Bernardo et al., 2013*; *De Nuccio et al., 2011*; *Roth et al., 2003*; *Saluja et al., 2001*) and inhibition with GW9662 blocks OL differentiation (*Bernardo et al., 2017*). Deficiency for the PPARγ-coactivator-1 alpha (PGC1a) leads to impaired lipid metabolism, including an increase in very long chain fatty acids (VLCFAs) and disruption of cholesterol homeostasis (*Camacho et al., 2013*; *Xiang et al., 2011*). In addition, PGC1a deficiency results in defects of peroxisome-related gene function, suggesting the increase in VLCFAs and drop in cholesterol reflects impaired peroxisome function (*Baes and Aubourg, 2009*). Following γ-secretase-dependent processing, the LRP1 ICD can translocate to the nucleus where it associates with transcriptional regulators (*Carter, 2007*; *May et al., 2002*). In endothelial cells, the LRP1-ICD binds directly to the nuclear receptor PPARγ to regulate gene products that function in lipid and glucose metabolism (*Mao et al., 2017*). Treatment of *Lrp1*[−/−] OPCs with pioglitazone leads to an increase in peroxisomes in OL processes but fails to promote differentiation into myelin sheet producing OLs. In the absence of the LRP1-ICD, pioglitazone may fail to fully activate PPARγ (*Mao et al., 2017*), but the observed increase in PMP70[+] peroxisomes in OL processes of *Lrp1* deficient cultures suggests that mutant cells still respond to pioglitazone. Because *Lrp1* cKO[OL] cultures are cholesterol deficient and the LRP1-ICD participates in PPARγ regulated gene expression, we examined whether a combination treatment rescues the differentiation block in *Lrp1* deficient OPCs/OLs. This was indeed the case, suggesting that *Lrp1* deficiency leads to dysregulation of multiple pathways important for OPC differentiation.

The importance of peroxisomes in the human nervous system is underscored by inherited disorders caused by complete or partial loss of peroxisome function, collectively described as Zellweger spectrum disorders (*Berger et al., 2016*; *Waterham et al., 2016*). *PEX* genes encode peroxins, proteins required for normal peroxisome assembly. Defects in *PEX* genes can cause peroxisome biogenesis disorder (PBD), characterized by a broad range of symptoms, including aberrant brain development, white matter abnormalities, and neurodegeneration (*Berger et al., 2016*). The genetic basis for PBD is a single gene mutation in one of the 14 PEX genes, typically leading to deficiencies in numerous metabolic functions carried out by peroxisomes (*Steinberg et al., 1993*). Mounting evidence points to a close interaction of peroxisomes with other organelles, mitochondria in particular, and disruption of these interactions may underlie the far reaching metabolic defects observed in PBD and genetically manipulated model organisms deficient for a single PEX (*Fransen et al., 2017*; *Wangler et al., 2017*). In developing OLs, *Lrp1* deficiency leads to a decrease in peroxisomal gene products, most prominently a ~50% reduction in PEX2, an integral membrane protein that functions in the import of peroxisomal matrix proteins. Mice deficient for *Pex2* lack normal peroxisomes but do assemble empty peroxisome membrane ghosts (*Faust and Hatten, 1997*). *Pex2* mutant mice show significantly lower plasma cholesterol levels and in the brain the rate of cholesterol synthesis is significantly reduced (*Faust and Kovacs, 2014*); brain size is reduced, cerebellar development impaired, and depending on the genetic background death occurs in early postnatal life (*Faust, 2003*). Mutations in human *PEX2* cause Zellweger spectrum disorder but have no apparent impact on white matter appearance (*Mignarri et al., 2012*). In mice, *CNP-Cre* mediated ablation of *Pex5* in OLs disrupts peroxisome function and integrity of myelinated fibers, but does not impair

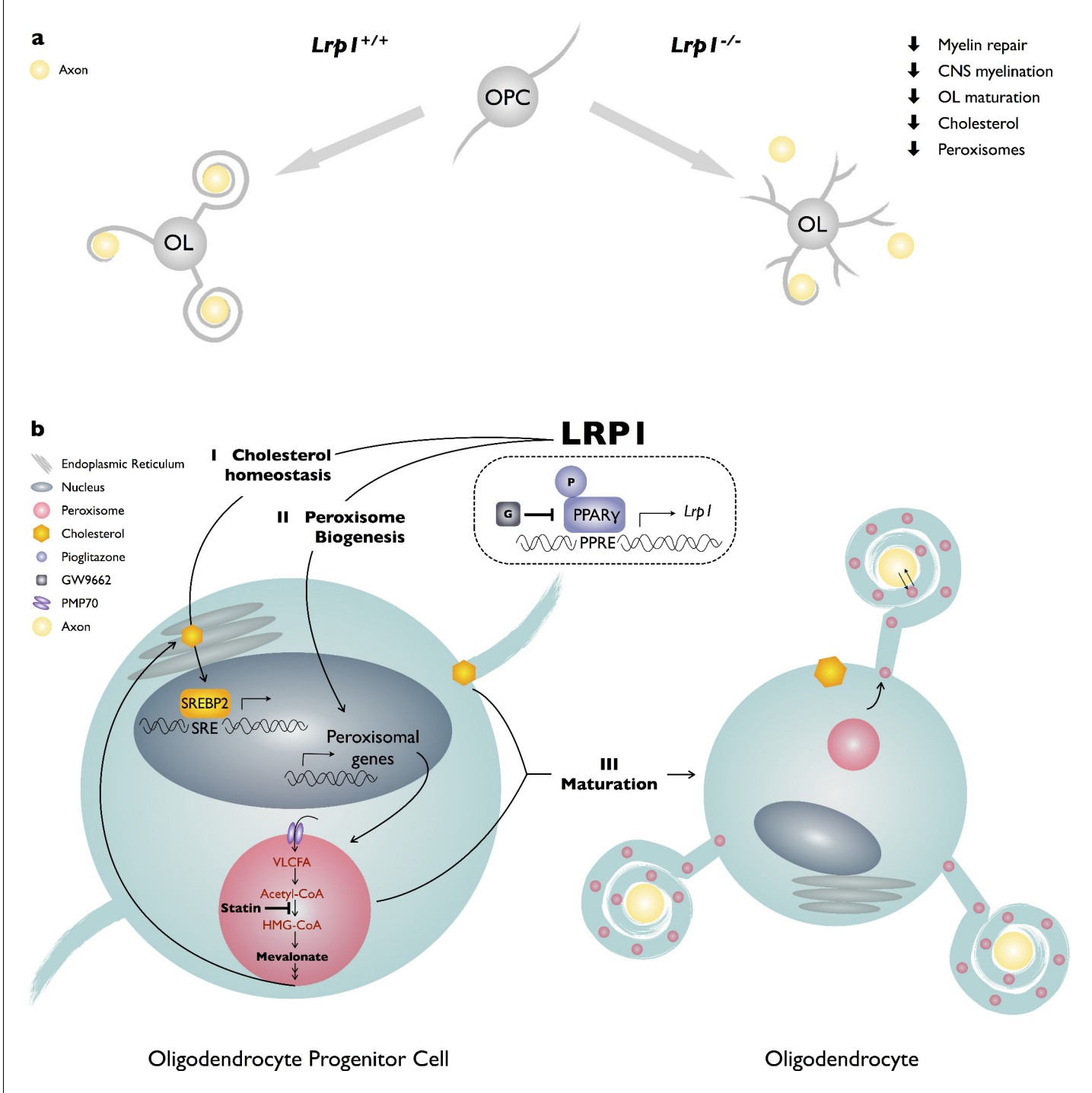

**Figure 8.** Working model of LRP1 regulated pathways in developing OLs. (**a**) LRP1 in the OL-lineage is necessary for proper CNS myelin development and the timely repair of a chemically induced focal white matter lesion. In OPCs, *Lrp1* deficiency leads to dysregulation of cholesterol homeostasis and impaired peroxisome biogenesis. (**b**) LRP1 is a key regulator of multiple pathways important for OPC differentiation into mature myelin producing OLs: (I) LRP1 regulates cholesterol homeostatsis; (II) LRP1 regulates peroxisome biogenesis; and (III) the combined treatment of *Lrp1* deficient primary OPCs with cholesterol and pioglitazone is sufficient to drive maturation into MBP$^+$ myelin sheet producing OLs.
DOI: https://doi.org/10.7554/eLife.30498.023

CNS myelinogenesis (*Kassmann et al., 2007*). This suggests that defects in CNS myelinogenesis observed in *Lrp1* cKO[OL] mice are likely not only a reflection of reduced peroxisome biogenesis or transport into internodes. Rather we provide evidence that *Lrp1* deficiency in OPCs leads to dysregulation of additional pathways implicated in myeliogenesis, including AKT, SREBP2, and PAPRγ. We propose that the combined action of these deficits attenuates OPC differentiation.

In sum, our studies show that *Lrp1* is required in the OL lineage for proper CNS myelin development and the timely repair of a chemically induced white matter lesion *in vivo*. Mechanistic studies with primary OPCs revealed that loss of *Lrp1* causes differentiation block that can be rescued by bath application of cholesterol combined with pharmacological activation of PPARγ.

# Materials and methods

**Key resources table**

| Reagent type (species) or resource | Designation | Source or reference | Identifiers | Additional information |
|---|---|---|---|---|
| Genetic reagent (*Mus musculus*) | *Lrp1[flox/flox]* | PMID:9634821 | RRID:IMSR_JAX:012604 | |
| Genetic reagent (*Mus musculus*) | *Olig2-Cre* | PMID:18691547 | RRID:MMRRC_011103-UCD | |
| Genetic reagent (*Mus musculus*) | *CAG-CreER[TM]* | PMID:11944939 | RRID:IMSR_JAX:004682 | |
| Genetic reagent (*Mus musculus*) | *Pdgfra-CreER[TM]* | PMID: 21092857 | RRID:IMSR_JAX:018280 | |
| Sequence-based reagent (cRNA) | *Pdgfra* cRNA | PMID:24948802 | | |
| Sequence-based reagent (cRNA) | *Plp1* cRNA | PMID:24948802 | | |
| Sequence-based reagent (cRNA) | *Mag* cRNA | PMID:22131434 | | |
| Sequence-based reagent (cRNA) | *Mbp* cRNA | this study | | Based on Allen Brain Atlas |
| Antibody | anti-Digoxigenin-AP antibody | Roche | #11093274910 | |
| Antibody | rabbit anti-Olig2 | Millipore | #AB9610, RRID:AB_570666 | |
| Antibody | rat anti-PDGFRα | BD Pharmingen | #558774, RRID:AB_397117 | |
| Antibody | rabbit anti-GFAP | DAKO | # A 0334, RRID:AB_10013482 | |
| Antibody | mouse anti-APC | Calbiochem | #OP80, Clone CC1, RRID:AB_2057371 | |
| Antibody | rabbit anti-Caspr | PMID: 9118959 | RRID:AB_2572297 | |
| Antibody | mouse anti-Na Channel | PMID: 10460258 | K58/35 | |
| Antibody | rabbit anti-CNPase | Aves Labs | #27490 R12-2096 | |
| Antibody | mouse anti-MAG | Millipore | #MAB1567, RRID:AB_2137847 | |
| Antibody | rat anti-MBP | Millipore | #MAB386, RRID:AB_94975 | |
| Antibody | chicken anti-PLP | Aves Labs | #27592 | |
| Antibody | mouse anti-GFAP | Sigma | #G3893, RRID:AB_477010 | |
| Antibody | chicken anti-GFAP | Aves Labs | #GFAP | |
| Antibody | rabbit anti-NG2 | Millipore | #AB5320, RRID:AB_91789 | |
| Antibody | rabbit anti-LRP1β | Abcam | #ab92544, RRID:AB_2234877 | |
| Antibody | rabbit anti-PMP70 | Thermo | #PA1-650, RRID:AB_2219912 | |
| Antibody | mouse anti-βIII tubulin | Promega | #G7121, RRID:AB_430874 | |
| Antibody | mouse anti-β-actin | Sigma | #AC-15 A5441, RRID:AB_476744 | |

*Continued on next page*

*Continued*

| Reagent type (species) or resource | Designation | Source or reference | Identifiers | Additional information |
|---|---|---|---|---|
| Antibody | rabbit anti-MAG | PMID: 27008179 | | |
| Antibody | rabbit anti-PLP | Abcam | #ab28486, RRID:AB_776593 | |
| Antibody | rat anti-PLP/DM20 | PMID: 27008179 | AA3 hybridoma | Wendy Macklin |
| Antibody | mouse anti-CNPase | Abcam | #ab6319, RRID:AB_2082593 | |
| Antibody | rabbit anti-PXMP3 (PEX2) | One world lab | #AP9179c | |
| Antibody | rabbit anti-SREBP2 | One world lab | #7855 | |
| Commercial assay or kit | Cholesterol/Cholesteryl Ester Quantitation Kit | Chemicon | #428901 | |
| Commercial assay or kit | DC™ Protein Assay | Bio-Rad | #5000112 | |
| Commercial assay or kit | RNeasy Micro Kit | Qiagen | #74004 | |
| Chemical compound, drug | 10 mM dNTP mix | Promega | #C1141 | |
| Chemical compound, drug | 5X Green GoTaq Buffer | Promega | #M791A | |
| Chemical compound, drug | GoTaq DNA polymerase | Promega | #M3005 | |
| Chemical compound, drug | L-$\alpha$-Lysophosphatidylcholine | Sigma | #L4129 | |
| Chemical compound, drug | Hoechst dye 33342 | Life technology | #H3570 | |
| Chemical compound, drug | ProLong Gold antifade reagent | Life technology | #P36930 | |
| Chemical compound, drug | Fluoromyelin-Green | Life technology | #F34651 | |
| Chemical compound, drug | PDGF-AA | Peprotech | #100-13A | |
| Chemical compound, drug | Forskolin | Sigma | #F6886 | |
| Chemical compound, drug | CNTF | Peprotech | #450–02 | |
| Chemical compound, drug | NT-3 | Peprotech | #450–03 | |
| Chemical compound, drug | T3 | Sigma | #T6397 | |
| Chemical compound, drug | Cholesterol | Sigma | #C8667 | |
| Chemical compound, drug | Pioglitazone | Sigma | #E6910 | |
| Chemical compound, drug | Simvastatin | Sigma | #S6196 | |
| Chemical compound, drug | GW9662 | Sigma | #M6191 | |
| Chemical compound, drug | Filipin | Sigma | #F9765 | |
| Chemical compound, drug | Super Signal West Pico substrate | Thermo | #34080 | |
| Chemical compound, drug | WesternSure PREMIUM Chemiluminescent Substrate | LI-COR Biosciences | #926–95000 | |
| Chemical compound, drug | Super Signal West Femto substrate | Thermo | #34095 | |
| Software, algorithm | Fiji | PMID: 22743772 | | |
| Software, algorithm | Axon pAlamp10.3 software | Molecular Devices | | |
| Software, algorithm | Origin9.1 software | Origin Lab | | |
| Software, algorithm | Image Studio Lite Western Blot Analysis Software | LI-COR Biosciences | | |
| Other | C-DiGit blot scanner | LI-COR Biosciences | #P/N 3600–00 | |
| Other | Multimode Plate Reader | Molecular Devices | #SpectraMax M5$^e$ | |
| Other | Stoelting stereotaxic instrument | Stoelting | #51730D | |
| Other | Stoelting quintessential stereotaxic injector | Stoelting | #53311 | |

## Mice

All animal handling and surgical procedures were performed in compliance with local and national animal care guidelines and were approved by the Institutional Animal Care and Use Committee (IACUC). $Lrp1^{flox/flox}$ mice were obtained from Steven Gonias (*Stiles et al., 2013*) and crossed with *Olig2-Cre* (*Schüller et al., 2008*), *CAG-CreER$^{TM}$* (Jackson Laboratories, #004682, Bar Harbor, ME), and *Pdgfra-CreER$^{TM}$* (*Kang et al., 2010*) mice. For inducible gene ablation in adult male and female mice, three intraperitoneal (i.p.) injections of tamoxifen (75 mg/kg) were given every 24 hr. Tamoxifen (10 mg/ml) was prepared in a mixture of 9% ethanol and 91% sunflower oil. Mice were kept on a mixed background of C57BL/6J and 129SV. Throughout the study, male and female littermate animals were used. *Lrp1* 'control' mice harbor at least one functional *Lrp1* allele. Any of the following genotypes $Lrp1^{+/+}$, $Lrp1^{+/flox}$, $Lrp1^{flox/flox}$, or $Lrp1^{flox/+};Cre^{+}$ served as *Lrp1* controls.

## Genotyping

To obtain genomic DNA (gDNA), tail biopsies were collected, boiled for 30 min in 100 µl alkaline lysis buffer (25 mM NaOH and 0.2 mM EDTA in ddH$_2$O) and neutralized by adding 100 µl of 40 mM Tris-HCl (pH 5.5). For PCR genotyping, 1–5 µl of gDNA was mixed with 0.5 µl of 10 mM dNTP mix (Promega, C1141, Madison, WI), 10 µl of 25 mM MgCl$_2$, 5 µl of 5X Green GoTaq Buffer (Promega, M791A), 0.2 µl of GoTaq DNA polymerase (Promega, M3005), 0.15 µl of each PCR primer stock (90 µM each), and ddH$_2$O was added to a total volume of 25 µl. The following cycling conditions were used: DNA denaturing step (94°C for 3 min) 1X, amplification steps (94°C for 30 s, 60°C for 1 min, and 72°C for 1 min) 30X, followed by an elongation step (72°C for 10 min) then kept at 4°C for storage. The position of PCR primers used for genotyping is shown in *Figure 1—figure supplement 1*. *Lrp1* WT and loxP-flanked (floxed) alleles were amplified with the forward primer [Lrp1tF10290, F2] 5'-CAT ACC CTC TTC AAA CCC TTT G-3' and the reverse primer [Lrp1tR10291, R2] 5'-GCA AGC TCT CCT GCT CAG ACC TGG A-3'. The WT allele yields a 291 bp product and the floxed allele yields a 350 bp product. The recombined *Lrp1* allele was amplified with the forward primer [Lrp1rF, F1] 5'- CCC AAG GAA ATC AGG CCT CGG C-3' and the reverse primer [R2], resulting in a 400 bp product (*Hennen et al., 2013*). For detection of *Cre*, the forward primer [oIMR1084, CreF] 5'-GCG GTC TGG CAG TAA AAA CTA TC-3' and reverse primer [oIMR1085, CreR] 5'-GTG AAA CAG CAT TGC TGT CAC TT-3' were used, resulting in a ~ 200 bp product. As a positive control, the forward primer [oIMR7338, Il-2pF] 5'-CTA GGC CAC AGA ATT GAA AGA-3' and the reverse primer [oIMR7339, Il-2pR] 5'-GTA GGT GGA AAT TCT AGC ATC-3' were mixed with CreF and CreR primers in the same reaction, this reaction yields a 324 bp product (The Jackson laboratory).

## Stereotaxic injection

Male and female mice at postnatal-day (P) 42–56 were used for stereotaxic injection of L-α-Lysophosphatidylcholine (LPC) (Sigma, L4129, Mendota Heights, MN) into the corpus callosum. Mice were anesthetized with 4% isoflurane mixed with oxygen, mounted on a Stoelting stereotaxic instrument (51730D, Wood Dale, IL), and kept under 2% isoflurane anesthesia during surgery. A 5µl-hamilton syringe was loaded with 1% LPC in PBS (Gibco, 10010023, Gaithersburg, MD), mounted on a motorized stereotaxic pump (Stoelting Quintessential Stereotaxic injector, 53311) and used for intracranial injection at the following coordinates, AP: 1.25 mm, LR:±1 mm, D: 2.25 mm. Over a duration of 1 min, 0.5 µl of 1% LPC solution was injected on the ipsilateral site and 0.5 µl PBS on the contralateral side. After the injection was completed, the needle was kept in place for 2 min before retraction. Following surgery, mice were treated with three doses of 70 µl of buprenorphine (0.3 mg/ml) every 12 hr. Brains were collected at day 10, and 21 post injection.

## Histochemistry

Animals were deeply anesthetized with a mixture of ketamine/xylazine (25 mg/ml ketamine and 2.5 mg/ml xylazine in PBS) and perfused trans-cardially with ice-cold PBS for 5 min, followed by ice-cold 4% paraformaldehyde in PBS (4%PFA/PBS) for 5 min. Brains were harvested and post-fixed for 2 hr in perfusion solution. Optic nerves were harvested separately and post-fixed for 20 min in perfusion solution. Brains and optic nerves were cryoprotected overnight in 30% sucrose/PBS at 4°C, embedded in OCT (Tissue-Tek, 4583, Torrance, CA), and flash frozen in powderized dry ice. Serial sections were cut at 20 µm (brains) and 10 µm (optic nerves) at −20°C using a Leica CM 3050S Cryostat.

Serial sections were mounted onto Superfrost[+] microscope slides (Fisherbrand, 12-550-15, Pittsburgh, PA) and stored at −20°C.

## In situ hybridization

Tissue sections mounted on microscope slides were post-fixed overnight in 4%PFA/PBS at 4°C. Sections were then rinsed 3 times for 5 min each in PBS and the edge of microscope slides was demarcated with a DAKO pen (DAKO, S2002, Denmark). Sections were subsequently incubated in a series of ethanol/water mixtures: 100% for 1 min, 100% for 1 min, 95% for 1 min, 70% for 1 min, and 50% for 1 min. Sections were then rinsed in 2x saline-sodium citrate (SSC, 150 mM NaCl, and 77.5 mM sodium citrate in ddH$_2$O, pH7.2) for 1 min, and incubated at 37°C for 30 min in proteinase K solution (10 µg/ml proteinase K, 100 mM Tris-HCl pH8.0, and 0.5 mM EDTA in ddH$_2$O). Proteinase digestion was stopped by rinsing sections in ddH$_2$O and then in PBS for 5 min each. To quench RNase activity, slides were incubated in 1% triethanolamine (Sigma, 90278) and 0.4% acetic anhydride (Sigma, 320102) mixture in ddH$_2$O for 10 min at room temperature, rinsed once in PBS for 5 min and once in 2X SSC for another 5 min. To reduce non-specific binding of cRNA probes, sections were pre-incubated with 125 µl hybridization buffer (10% Denhardts solution, 40 mg/ml baker's yeast tRNA, 5 mg/ml sheared herring sperm DNA, 5X SSC, and 50% formamide in ddH$_2$O) for at least 2 hr at room temperature. Digoxigenin-labeled cRNA probes were generated by run-off *in vitro* transcription as described (*Winters et al., 2011*). Anti-sense and sense cRNA probes were diluted in 125 µl pre-hybridization buffer to ∼200 ng/ml, denatured for 5 min at 85°C, and rapidly cooled on ice for 2 min. Probes were applied to tissue sections, microscope slides covered with parafilm, and incubated at 55°C overnight in a humidified and sealed container. The next morning slides were rinsed in 5X SSC for 1 min at 55°C, 2X SSC for 5 mins at 55°C, and incubated in 0.2X SSC/50% formamide for 30 min at 55°C. Sections were rinsed in 0.2X SSC at room temperature for 5 min then rinsed with Buffer1 (100 mM Tris-HCl pH7.5, and 1.5M NaCl in ddH$_2$O) for 5 min. A 1% blocking solution was prepared by dissolving 1 g blocking powder (Roche, 11096176001, Switzerland) in Buffer1 at 55°C, cooled to room temperature (RT), and applied to slides for 1 hr at RT. Slides were rinsed in Buffer1 for 5 min and 125 µl anti-Digoxigenin-AP antibody (Roche, 11093274910, 1:2500) in Buffer1 was applied to each slide for 1.5 hr at RT. Sections were rinsed in Buffer1 for 5 min, then rinsed in Buffer2 (100 mM Tris-HCl pH9.5, 100 mM NaCl, and 5 mM MgCl$_2$ in ddH$_2$O) for 5 min, and incubated in alkaline phosphatase (AP) substrate (Roche, 11681451001, 1:50) in Buffer2. The color reaction was developed for 1–48 hr and stopped by rinsing sections in PBS for 10 min. Sections were incubated in Hoechst dye 33342 (Life technology, H3570, Pittsburgh, PA) for 5 min, air dried, mounted with Fluoromount-G (SouthernBiotech, 0100–01), and dried overnight before imaging under bright-field. The following cRNA probes were used, *Pdgfra* and *Plp* (DNA templates were kindly provided by Richard Lu (*Dai et al., 2014*)), *Mag* (*Winters et al., 2011*), and *Mbp* (a 650 bp probe based on template provided in the Allen Brain Atlas).

## Quantification of lesion size and myelin repair

Serial sections of the corpus callosum, containing the LPC and PBS injection sites were mounted onto glass coverslips and stained by ISH with digoxigenin-labeled cRNA probes specific for *Mbp*, *Mag, Plp*, and *Pdgfra*. For quantification of the white matter lesion area, the same intensity cutoff was set by Image J threshold for all brain sections and used to measure the size of the lesion. The outer rim of the strongly *Mbp*[+] region (lesion[out]) was traced with the ImageJ freehand drawing tool. The inner rim facing the *Mbp*[-] region (lesion[in]) was traced as well. For each animal examined, the size of the initial lesion area (lesion[out]) in µm$^2$ and remyelinated area (lesion[out]-lesion[in]) in µm$^2$ was calculated by averaging the measurement from two sections at the lesion core. The lesion core was defined as the section with the largest lesion area (lesion[out]). To determine remyelination, the ratio of (lesion[out]-lesion[in])/(lesion[out]) in percent was calculated. As an initial lesion depth control, criteria of lesion[out] area must cover the center of the corpus callosum in each serial section set. If a lesion[out] area was not located within the corpus callosum, the animal and corresponding brain sections were excluded from the analysis.

## Immunostaining

Tissue sections mounted onto microscope slides were rehydrated in PBS for 5 min, permeabilized in 0.1% TritonX-100, and blocked in PHT (1% horse serum and 0.1% TritonX-100 in PBS) for 1 hr at RT. Primary antibodies were diluted in PHT and applied overnight at 4°C. Sections were rinsed in PBS 3 times for 5 min each and appropriate secondary antibodies were applied (Life technologies, Alexa-fluorophore 405, 488, 555, 594, or 647 nm, 1:1000). Slides were rinsed in PBS 3 times for 5 min each and mounted with ProLong Gold antifade reagent (Life technologies, P36930). For quantification of nodal structures, randomly selected fields of view in each nerve were imaged at 96X magnification with an Olympus IX71 microscope, a maximum projection of 6 Z-stacked images of each region was generated, and the stacked images were used for quantification. As axons run in and out of the plane within longitudinal sections, criteria were set to exclude structures in which Caspr staining was unpaired to reduce 'false positive' as nodal defect. The following primary antibodies were used: rabbit anti-Olig2 (Millipore, AB9610, Burlington, MA, 1:500), rat anti-PDGFRα (BD Pharmingen, 558774, San Jose, CA, 1:500), rabbit anti-GFAP (Dako, Nr. A 0334, 1:2000), mouse anti-APC (Calbiochem, OP80, Clone CC1, San Diego, CA, 1:500), rabbit anti-Caspr (1:1000, [*Peles et al., 1997*]), mouse anti-Na Channel (1:75, [*Rasband et al., 1999*]). For myelin staining, sections were incubated in Fluoromyelin-Green (Life technologies, F34651 1:200) reagent for 15 min.

## Transmission electron microscopy (TEM)

Tissue preparation and image acquisition were carried out as described by *Winters et al. (2011)*. Briefly, mice at P10, P21, and P56 were perfused trans-cardially with ice cold PBS for 1 min, followed by a 10 min perfusion with a mixture of 3% PFA and 2.5% glutaraldehyde in 0.1M Sorensen's buffer. Brains and optic nerves were dissected and post-fixed in perfusion solution overnight at 4°C. Post-fixed brain tissue and optic nerves were rinsed and transferred to 0.1M Sorensen's buffer and embedded in resin by the University of Michigan Imaging Laboratory Core. Semi-thin (0.5 μm) sections were cut and stained with toluidine blue and imaged by light microscopy. Ultra-thin (75 nm) sections were cut and imaged with a Philips CM-100 or a JEOL 100CX electron microscope. For each genotype and age, at least three animals were processed and analyzed. For each animal, over 1000 axons in the optic nerve were measured and quantified by ImageJ. For each optic nerve, 10 images at 13,500x magnification were randomly taken and quantified to calculate the g-ratio and the fraction of myelinated axons. The inner ($area^{in}$) and outer ($area^{out}$) rim of each myelin sheath was traced with the ImageJ freehand drawing tool and the area within was calculated. We then derived axon caliber and fiber caliber (2 r) by the following: $area^{in} = r^2\pi$. The g-ratios were calculated as such: $\frac{\sqrt{area^{in}}}{\sqrt{area^{out}}}$. The g-ratio is only accurate if the compact myelin and axon outline can clearly be traced. Individual fibers with not clearly defined features were excluded from the quantification.

## Optic nerve recordings

Compound action potentials were recorded as described elsewhere (*Carbajal et al., 2015*; *Winters et al., 2011*). Briefly, optic nerves were acutely isolated from P21 mice and transferred into oxygenated ACSF buffer (125 mM NaCl, 1.25 mM $NaH_2PO_4$, 25 mM glucose, 25 mM $NaHCO_3$, 2.5 mM $CaCl_2$, 1.3 mM $MgCl_2$, 2.5 mM KCl) for 45 min at RT before transferring into a recording chamber at $37 \pm 0.4$°C. Suction pipette electrodes were used for stimulation and recording of the nerve (*Figure 2—figure supplement 2a*). A computer-driven (Axon pClamp10.3 software) stimulus isolation unit (WPI, FL) was used to stimulate the optic nerve with 2 mA/50 μs pulses. The recording electrode was connected to a differential AC amplifier (custom-made). A stimulus artifact-subtracting pipette was placed near the recording pipette. A data acquisition system (Axon digidata 1440A, Axon pClamp 10.3, Molecular Devices, Sunnyvale, CA) was used to digitize the signals. Conduction velocity was calculated from the length of the nerve and the time to peak of each component of the CAP. Amplitudes were normalized to a resistance ratio of 1.7, as described (*Fernandes et al., 2014*). Raw traces were fitted with four Gaussian curves with Origin9.1 software for analysis of individual components of the CAP. Due to limitations in the resolution of individual peaks in short nerves, CAP recordings from nerves that were shorter than 1 mm in length were excluded from the analysis.

## OPC/OL primary cultures and drug treatment

OPCs were isolated from P6-P9 mouse pups by rat anti-PDGFRα (BD Pharmingen, 558774) immuno-panning as described (*Mironova et al., 2016*). For plating of cells, $5-7.5 \times 10^3$ cells (for 12 mm cover glass) or $3-5 \times 10^4$ (12-well plastic plate) were seeded onto PDL pre-coated surface. Primary OPCs were kept in a 10% $CO_2$ incubator at 37°C. To maintain OPCs in a proliferative state, growth medium (20 ng/ml PDGF-AA (Peprotech, 100-13A, Rocky Hill, NJ), 4.2 µg/ml Forskolin (Sigma, F6886), 10 ng/ml CNTF (Peprotech, 450–02), and 1 ng/ml NT-3 (Peprotech, 450–03) in SATO) were added to the culture. To induce OPC differentiation, differentiation medium was constituted by adding (4.2 µg/ml Forskolin, 10 ng/ml CNTF, and 4 ng/ml T3 (Sigma, T6397) in SATO) to the culture. For drug treatment, all compounds were mixed with differentiation medium at the desired concentration, and the compound-containing medium was replaced every other day. Stock and working solutions including 20 mg/ml cholesterol (Sigma, C8667) in 100% EtOH were kept at RT and warmed up to 37°C before use, then diluted in differentiation medium to 5 µg/ml; 10 mM pioglitazone (Sigma, E6910) in DMSO was kept at −20°C and diluted in differentiation medium to 1 µM; 10 mM simvastatin (Sigma, S6196) in DMSO was kept at −20°C and diluted in differentiation medium to 0.5 µM; 10 mM GW9662 (Sigma, M6191) in DMSO was kept at −20°C and diluted in differentiation medium to 1 µM.

## OPC staining and quantification

At different stages of development, OPC/OL cultures were fixed for 15 min in 4%PFA/PBS. Cells were rinsed three times in PBS and permeabilized with 0.1% Triton/PBS solution for three mins. Cells were then rinsed in PBS and incubated in blocking solution (3% BSA/PBS) for 1 hr at RT. Primary anti-bodies were prepared in blocking solution. For immunostaining, 35 µl were dropped onto a sheet of parafilm, the coverslips were inverted onto the primary antibody drop, and incubated overnight at 4°C. The following day, coverslips were transferred back to a 24-well-plate and rinsed with PBS 3 times for 5 min each. Secondary antibody ±filipin (Sigma, F9765, 0.1 mg/ml) was prepared in blocking solution, 350 µl were added to each well, and the coverslips were incubated for 2 hr at RT. Coverslips then were rinsed in PBS three times for 5 min each and stained with Hoechst (1:50,000) for 10 s. Coverslips then were rinsed in ddH$_2$O and mounted in ProLong Gold antifade reagent. For quantification in *Figure 3*, the percent of OL markers$^+$/Hoechst$^+$ cells was calculated from 10 images that were taken from randomly selected areas in each coverslip at 20X magnification with an Olympus IX71 microscope. For quantification in *Figure 4* and after, the percent of OL markers+/Hoechst + cells was calculated from 25 images that were taken from randomly selected areas in each coverslip at 10X magnification with a Zeiss Axio-Observer microscope. For single-cell intensity and size measurement in *Figures 4–7*, individual cell images were taken at 40X magnification with a Zeiss Axio-Observer microscope with Apotom.2. For quantification, the same intensity cutoff was set by Image J threshold to all cells and binary images were generated to define each cell outline. The individual cell outline was applied to original images to measure the intensity of filipin, MBP, or PMP70 staining per cell. For PMP70 puncta distribution analysis, the coordinates of each PMP70$^+$ center were acquired by the process >find maxima function in ImageJ, the cell center coordinate was defined by point selection function, and the distance of each PMP70$^+$ dot to the cell center was then calculated. The data were then binned from 1 to 25 µm at 1 µm divisions, and plotted. Primary antibodies included: rat anti-PDGFRα (BD pharmingen, 558774, 1:500), rabbit anti-CNPase (Aves Labs, 27490 R12-2096, Tigard, OR, 1:500), mouse anti-MAG (Millipore, MAB1567, 1:500), rat anti-MBP (Millipore, MAB386, 1:1000), chicken anti-PLP (Aves Labs, 27592, 1:500), mouse anti-GFAP (Sigma, G3893, 1:1000), chicken anti-GFAP (Aves Labs, GFAP, 1:500), rabbit anti-NG2 (Millipore, AB5320, 1:500), rabbit anti-LRP1β (Abcam, ab92544, 1:500), rabbit anti-PMP70 (Thermo, PA1-650, Waltham, MA, 1:1000).

## Western blot analysis

Protein lysates were separated by SDS-PAGE and transferred onto PVDF membranes for immuno-blotting. Depending on the application, 2 to 10 µg of total protein were loaded per well. 2% Blotting-Grade Blocker (Bio-Rad, #170–6404, Hercules, CA) or 2% BSA fraction V (Fisher, BP1600-100) in 0.1%TBST buffer (0.1% Tween-20, 3M NaCl, 200 mM Tris-HCl pH7.4) were used as blocking solutions and membranes were incubated for 1 hr at RT. Primary antibodies were diluted in blocking

buffer and used for incubation at 4°C overnight. For protein detection and densitometric analysis, membranes were incubated in Super Signal West Pico substrate (Thermo, 34080), WesternSure PRE-MIUM Chemiluminescent Substrate (LI-COR Biosciences, 926–95000, Lincoln, NE), or Super Signal West Femto substrate (Thermo, 34095) followed by scanning on a C-DiGit blot scanner (LI-COR, P/N 3600–00). Images were quantified with Image Studio Lite Western Blot Analysis Software, relative to loading controls. Blots were used for quantification only when the loading control signals were comparable between groups and signals between technical repeats were similar. Primary antibodies included: rabbit anti-LRP1β 85 kDa (Abcam, ab92544, United Kingdom, 1:2000), mouse anti-βIII tubulin (Promega, G7121, 1:5000), mouse anti-β-actin (Sigma, AC-15 A5441, 1:5000), rat anti-MBP (Millipore, MAB386, 1:1000), rabbit anti-MAG (homemade serum, 1:1000), rabbit anti-PLP (Abcam, ab28486, 1:1000), rat anti-PLP/DM20 (Wendy Macklin AA3 hybridoma, 1:500) rabbit anti-Olig2 (Millipore, AB9610, 1:1000), mouse anti-GFAP (Sigma, G3893, 1:1000), mouse anti-CNPase (Abcam, ab6319, 1:1000), rabbit anti-PXMP3 (PEX2) (One world lab, AP9179c, San Diego, CA, 1:250), and rabbit anti-SREBP2 (One world lab, 7855, 1:500).

## Cholesterol measurement

OPCs were isolated by immunopanning as described above. OPCs bound to panning plates were collected by scraping with a Scraper (TPP, TP99002) in 250 µl of ice-cold PBS and sonicated in an ice-cold water bath (Sonic Dismembrator, Fisher Scientific, Model 500) at 50% amplitude three times for 5 s with a 5-s interval. The sonicated cell suspensions were immediately used for cholesterol measurement following the manufacturer's instructions (Chemicon, 428901). For colorimetric detection and quantification of cholesterol, absorbance was measured at 570 nm with a Multimode Plate Reader (Molecular Devices, SpectraMax M5$^e$). Results were normalized to total protein concentration measured by DC$^{TM}$ Protein Assay according to the manufacturer's manual (Bio-Rad, 5000112).

## Microarray and gene ontology analysis

OPCs were isolated by immunopanning as described above and RNA was isolated with the RNeasy Micro Kit (Qiagen, 74004, Germany). To compare *Lrp1* control and cKO$^{OL}$ RNA expression profiles, the Mouse Gene ST2.1 Affymetrix array was used. Differentially expressed genes, with a p-value<0.05 set as cutoff, were subjected to gene ontology (GO) analysis. Go terms were quarried from Mouse Genome Informatics (MGI) GO browser. The fold enrichment was calculated by dividing the number of genes associated with the GO term in our list by the number of genes associated with the GO term in the database.

## Statistical analysis

There was no pre-experimental prediction of the difference between control and experimental groups when the study was designed. Therefore, we did not use computational methods to determine sample size a priori. Instead, we use the minimum of mice per genotype and experimental treatment for a total of at least three independent experiments to achieve the statistical power discussed by *Gauch, 2006*). We used littermate *Lrp1* control or *Lrp1* cKO or iKO mice for comparison throughout the study. All independent replicas were biological replicas, rather than technical replicas. For each experiment, the sample size (n) is specified in the figure legend. Throughout the study, independent replicas (n) indicate biological replica. Technical replicas were used to control for the quality of each measurement and were averaged before quantification and the average value was used as (n = 1) biological replica. Unless indicated otherwise, results are represented as mean value ±SEM. For single pairwise comparison, Student's *t*-test was used and a p-value<0.05 was considered statistically significant. For multiple comparisons, two-way ANOVA followed by post hoc *t*-test were used. Numbers and R software (see source code file for details) were used for determining statistical significance and graph plotting. For detailed raw data and statistical report, see source data files for each figure. For image processing and quantification, ImageJ 1.47 v software was used for threshold setting, annotation, and quantification.

## Acknowledgements

We thank Andy Lieberman for providing *CAG-CreER$^{TM}$* mice, Ben Barres for providing *Olig2-Cre* mice, and Richard Lu for *Plp1* plasmid DNA. We thank Chang-Ting Lin, Wei-Chin Ho, and Chih-Hsu

Lin for bioinformatics consulting. This work was supported by a Bradley Merrill Patten Fellowship (J-PL), the Training Program in Organogenesis T32HD007505 and NIH Cellular and Molecular Biology Training Grant T32-GM007315 (YAM), R01 NS081281 (PS and RJG), the Schmitt Program on Integrative Brain Research (PS), and the Dr. Miriam and Sheldon G. Adelson Medical Foundation on Neural Repair and Rehabilitation (RJG).

## Additional information

### Funding

| Funder | Grant reference number | Author |
| --- | --- | --- |
| Eunice Kennedy Shriver National Institute of Child Health and Human Development | T32HD007505 | Yevgeniya A Mironova |
| National Institute of General Medical Sciences | T32GM007315 | Yevgeniya A Mironova |
| National Institute of Neurological Disorders and Stroke | R01NS081281 | Peter Shrager Roman J Giger |
| Schmitt Program on Integrative Brain Research | | Peter Shrager |
| Dr. Miriam and Sheldon G. Adelson Medical Research Foundation | APNRR | Roman J Giger |
| Bradley Merrill Patten Fellowship | | Jing-Ping Lin |

The funders had no role in study design, data collection and interpretation, or the decision to submit the work for publication.

### Author contributions

Jing-Ping Lin, Conceptualization, Resources, Data curation, Formal analysis, Validation, Investigation, Visualization, Methodology, Writing—original draft, Project administration, Writing—review and editing; Yevgeniya A Mironova, Conceptualization, Supervision, Validation, Methodology, Project administration, Writing—review and editing; Peter Shrager, Conceptualization, Resources, Formal analysis, Supervision, Funding acquisition, Validation, Methodology, Project administration, Writing—review and editing; Roman J Giger, Conceptualization, Resources, Supervision, Funding acquisition, Visualization, Writing—original draft, Project administration, Writing—review and editing

### Author ORCIDs

Jing-Ping Lin http://orcid.org/0000-0003-0686-0215
Roman J Giger http://orcid.org/0000-0002-2926-3336

### Ethics

Animal experimentation: This study was performed in strict accordance with the recommendations in the Guide for the Care and Use of Laboratory Animals of the National Institutes of Health. All of the animals were handled according to protocols approved by the University committee on use and care for animals (IACUC protocols: #00005863 and #00005896) of the University of Michigan.

### Decision letter and Author response

Decision letter https://doi.org/10.7554/eLife.30498.027
Author response https://doi.org/10.7554/eLife.30498.028

## Additional files

### Supplementary files

• Source code 1. ANOVA source code.
DOI: https://doi.org/10.7554/eLife.30498.024

• Transparent reporting form
DOI: https://doi.org/10.7554/eLife.30498.025

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
