## [Decision Letter]

Thank you for submitting your article "LRP1 Regulates Peroxisome Biogenesis and Cholesterol Homeostasis in Oligodendrocytes and is Required for CNS Myelin Development and Repair" for consideration by *eLife*. Your article has been favorably evaluated by Didier Stainier (Senior Editor) and three reviewers, one of whom is a member of our Board of Reviewing Editors. The reviewers have opted to remain anonymous.

The reviewers have discussed the reviews with one another and the Reviewing Editor has drafted this decision to help you prepare a revised submission.

Summary:

Lin and colleagues have explored the role of low-density lipoprotein related-receptor-1 (LRP1) in CNS myelination. They propose that LRP1 has two related functions in cholesterol homeostasis and peroxisome formation in oligodendrocytes (OLs) during myelination. Specifically, they find that LRP1 knockout mice have a modestly reduced number of myelinated axons in the optic nerve, reduced nerve conduction as quantified by ex vivo optic nerve compound action potentials, and exhibit slowed OL differentiation. They can show that one cause of the differentiation block is a defect in reaching normal cholesterol levels, presumably required for myelin protein stabilization and myelin membrane synthesis. The addition of cholesterol to a culture medium has little rescuing effect, but this improves with adding also pioglitazone (a PPARγ agonist), an experiment prompted by the finding of dysregulated peroxisomal genes. The authors conclude a new function of LRP1 in myelination with a novel "link" between cholesterol and the role of peroxisomes.

The principal observation of the authors is very interesting and they have undoubtedly found evidence for a previously unrecognized role of LRP1 in oligodendrocyte differentiation and myelination. Overall, the data are new, robust and compelling. However, there are a few major points to be addressed in a revised version.

Major points:

1) One major concern is that the observed phenotype in conditional KO mice may not entirely be due to loss of LRP1. The authors do not specify the genotypes of mice being used as controls for each experiment. This is important because *Olig2-Cre* is a knockin allele that ablates the endogenous Olig2 gene (Schuller et al. 2008 Cancer Cell), and it has been shown that Olig2 heterozygotes exhibit delays in OL differentiation *in vivo* (e.g. Liu et al. Dev. Bio 302, 2007). Therefore it is possible that the modest delays in OL differentiation leading to mild myelination defects results from comparing *Olig2* heterozygotes (*Lrp1^flox/flox^; Olig2-Cre/+*) to wild-type mice, and that LRP1 is not contributing much to this phenotype. Extending this, reduced cholesterol level in mutant OLs could be a sequella of slower differentiation. Thus the specific genotype of control mice must be assessed for each experiment.

2) The result in Figure 1—figure supplement 1.e. tamoxifen-treated CAG-CreERT2/Lrp1-floxed mice, appears not essential because it is unclear whether the CAG transgene is expressed in all cells, as the Western blot data suggest this is not the case. It is also unclear whether dysmyelination (which appears very heterogeneous by EM) stems from mosaic Cre expression in the OPC (possibly with delayed recombination also in myelinating OL) and/or what neuronal and astroglial LRP1 deficiency contributes to dysmyelination. The size distribution of myelinated axons is not normal and that could be cause of consequence of dysmyelination. With these uncertainties, one cannot conclude anything and these mutants are also of little help for the problem associated with Olig2 heterozygosity. Shortening the manuscript would also help the readability.

3) The observation of reduced peroxisome numbers and their altered localization is interesting, but whether this is causal to the myelination defect or by itself part of a differentiation block has not be answered. The "promyelinating" function of the PPAR agonist is not reflected in peroxisomal biogenesis and more likely a result of improved (peroxisome-independent) lipid metabolism. The authors did not discuss relevant prior research on peroxisome biogenesis and myelination (e.g. Kassmann et al., Nat Genet 2007), which demonstrates that peroxisomes are completely dispensable for oligodendrocyte differentiation, but essential for maintaining adult white matter tracts. Zellweger syndrome is a severe developmental disease due to the role of peroxisomes in cells other than oligodendrocytes. Adrenoleukodystrophy is an early onset demyelinating disease. Figure 7 concludes a role of peroxisomes in OPC differentiation that has no experimental basis yet.

4) The effects of cholesterol and PPAR agonist, when combined, on the *in vitro* differentiation of LRP1-deficient OL is significant. However, this observation identifies no specific interaction of these pathways (Discussion: "a novel link") other than the requirement of lipid synthesis for myelination, which is well known. Other equally important aspects of the pleitropic effects of LRP11 deficiency on myelination may exist.

---

## [Author Response]

Major points:1) One major concern is that the observed phenotype in conditional KO mice may not entirely be due to loss of LRP1. The authors do not specify the genotypes of mice being used as controls for each experiment. This is important because Olig2-Cre is a knockin allele that ablates the endogenous Olig2 gene (Schuller et al. 2008 Cancer Cell), and it has been shown that Olig2 heterozygotes exhibit delays in OL differentiation in vivo (e.g. Liu et al. Dev. Bio 302, 2007). Therefore it is possible that the modest delays in OL differentiation leading to mild myelination defects results from comparing Olig2 heterozygotes (Lrp1^flox/flox^; Olig2-Cre/+) to wild-type mice, and that LRP1 is not contributing much to this phenotype. Extending this, reduced cholesterol level in mutant OLs could be a sequella of slower differentiation. Thus the specific genotype of control mice must be assessed for each experiment.

We are aware of the studies by (Schuller et al., 2008) and (Liu et al., 2007) and agree with the reviewers that loss of one allele of *Olig2* may have potential confounding effects on CNS myelin development. To address this point, we ran additional Western blots of P21 brains obtained from *Lrp1^flox/+^;Olig2-Cre* and *Lrp1^flox/+^* mice (n= 3 mice per genotype) and probed the blots with antibodies specific for MBP, MAG, CNP and LRP1. As shown in Figure 2—figure supplement 1 and 1C of the revised manuscript, mice heterozygous for *Olig2* show protein levels comparable to mice wildtype for *Olig2*. To address whether primary OPCs prepared from *Lrp1^flox/+^;Olig2-Cre* and *Lrp1^flox/+^* pups show differences in LRP1β, MAG, or PLP we ran additional WBs and found no significant differences (Figure 3—figure supplement 1). Moreover, primary OPC/OL cultures prepared from *Lrp1^flox/+^;Olig2-Cre* and *Lrp1^flox/+^* pups, show comparable levels of the cholesterol sensor SREBP2 (Figure 4—figure supplement 1). In sum, additional control experiments shown in the revised manuscript, confirm and strengthen our conclusions that *Lrp1* is important for OPC differentiation *in vitro* and proper CNS myelination *in vivo*.

2) The result in Figure 1—figure supplement 1.e. tamoxifen-treated CAG-CreERT2/Lrp1-floxed mice, appears not essential because it is unclear whether the CAG transgene is expressed in all cells, as the Western blot data suggest this is not the case. It is also unclear whether dysmyelination (which appears very heterogeneous by EM) stems from mosaic Cre expression in the OPC (possibly with delayed recombination also in myelinating OL) and/or what neuronal and astroglial LRP1 deficiency contributes to dysmyelination. The size distribution of myelinated axons is not normal and that could be cause of consequence of dysmyelination. With these uncertainties, one cannot conclude anything and these mutants are also of little help for the problem associated with Olig2 heterozygosity. Shortening the manuscript would also help the readability.

We appreciate the reviewers’ comments and agree that we do not know whether any of the phenotypes observed in *Lrp1* inducible knock-out (iKO) mice is due to loss of *Lrp1* in OPC/OLs, neurons, astrocytes, microglia or other cell types. The reviewers are also correct that tamoxifen induced *Lrp1* ablation is incomplete, resulting in a mosaic mouse. Several years ago, when we embarked on this project, we first carried out studies with *Lrp1* iKO mice – based on defects in CNS myelin development and white matter repair we concluded that *Lrp1* is important for proper CNS myelination – in subsequent experiments we refined our approach using more specific Cre-driver lines (i.e. *Olig2-Cre* and *PDGFRα-CreER* lines). Given the hypomyelination phenotype observed in *Lrp1^flox/flox^;Olig2-Cre* mice, experiments with *Lrp1 iKO* mice may no longer be relevant to demonstrate a role for normal CNS myelin. As suggested by the reviewers, we have removed developmental studies with *Lrp1 iKO* mice to shorten the manuscript.

For the reasons listed below, we think there is value in presenting the white matter repair studies in LPC injected adult *Lrp1 iKO* mice:

*- Lrp1* is required for proper CNS myelin development, and thus, inducible gene ablation in adult mice (after myelination is completed) is needed to study white matter repair. Any iKO-based approach will have some limitations, including incomplete recombination in the cell type of interest, leading to “mosaic” mice.

- Though inducible gene ablation is not very efficient and clearly incomplete in *Lrp1^flox/flox^;CAG-CreER* mice, the delay in white matter repair in iKO mice demonstrates for the first time that *Lrp1* function is required for the timely repair of damaged adult CNS white matter. A finding that prompted cell type specific gene ablation studies with *PDGFRα-CreER* mice. Inducible ablation of *Lrp1* is the OL lineage in tamoxifen treated adult *Lrp1^flox/flox^;PDGFRα-CreER* mice revealed a delay in CNS white matter repair, expanding and refining our original observation in *Lrp1^flox/flox^;CAG-CreER* mice.

3) The observation of reduced peroxisome numbers and their altered localization is interesting, but whether this is causal to the myelination defect or by itself part of a differentiation block has not be answered. The "promyelinating" function of the PPAR agonist is not reflected in peroxisomal biogenesis and more likely a result of improved (peroxisome-independent) lipid metabolism. The authors did not discuss relevant prior research on peroxisome biogenesis and myelination (e.g. Kassmann et al., Nat Genet 2007), which demonstrates that peroxisomes are completely dispensable for oligodendrocyte differentiation, but essential for maintaining adult white matter tracts. Zellweger syndrome is a severe developmental disease due to the role of peroxisomes in cells other than oligodendrocytes. Adrenoleukodystrophy is an early onset demyelinating disease. Figure 7 concludes a role of peroxisomes in OPC differentiation that has no experimental basis yet.

In the revised manuscript we discuss previous work on PEX5 and the role of peroxisomes in white matter stability e.g. (Kassmann et al., 2007). We agree with the reviewer that peroxisome deficiency alone does not impair CNS myelin development. However, since *Lrp1* ablation leads to (i) a reduction in peroxisomal gene products, (ii) altered PPARγ signaling and, (iii) dysregulation of cholesterol homeostasis, the combined effects of multiple metabolic defects is a likely cause for impaired OPC differentiation. To make this clear we have revised the Discussion accordingly:

“In developing OLs, *Lrp1* deficiency leads to a decrease in peroxisomal gene products, most prominently a ~50% reduction in PEX2, an integral membrane protein that functions in the import of peroxisomal matrix proteins. […] We propose that the combined action of these deficits attenuates OPC differentiation.”

4) The effects of cholesterol and PPAR agonist, when combined, on the in vitro differentiation of LRP1-deficient OL is significant. However, this observation identifies no specific interaction of these pathways (Discussion: "a novel link") other than the requirement of lipid synthesis for myelination, which is well known. Other equally important aspects of the pleitropic effects of LRP11 deficiency on myelination may exist.

We appreciate the reviewers’ comments and provide clarification on this point – e.g. the link we refer to is a “functional link” and not a “mechanistic link” – we may not have made this clear in the original version of the manuscript. The Discussion of the revised manuscript reads as follows:

“Taken together, our studies identify a novel role for LRP1 in peroxisome function and suggest that broad metabolic dysregulation in *Lrp1* deficient OPCs attenuates differentiation into mature OLs (Figure 8).”